# Mechanics and kinetics of dynamic instability

**Thomas CT Michaels[1], Shuo Feng[2,3], Haiyi Liang[2,3]\*, L Mahadevan[1,4,5]\***

[1]Paulson School of Engineering and Applied Sciences, Harvard University, Cambridge, United States; [2]Department of Modern Mechanics, University of Science and Technology of China, Hefei, China; [3]IAT Chungu Joint Laboratory for Additive Manufacturing, Anhui Chungu 3D Institute of Intelligent Equipment and Industrial Technology, Wuhu, China; [4]Department of Physics, Harvard University, Cambridge, United States; [5]Department of Organismic and Evolutionary Biology, Harvard University, Cambridge, United States

**Abstract** During dynamic instability, self-assembling microtubules (MTs) stochastically alternate between phases of growth and shrinkage. This process is driven by the presence of two distinct states of MT subunits, GTP- and GDP-bound tubulin dimers, that have different structural properties. Here, we use a combination of analysis and computer simulations to study the mechanical and kinetic regulation of dynamic instability in three-dimensional (3D) self-assembling MTs. Our model quantifies how the 3D structure and kinetics of the distinct states of tubulin dimers determine the mechanical stability of MTs. We further show that dynamic instability is influenced by the presence of quenched disorder in the state of the tubulin subunit as reflected in the fraction of non-hydrolysed tubulin. Our results connect the 3D geometry, kinetics and statistical mechanics of these tubular assemblies within a single framework, and may be applicable to other self-assembled systems where these same processes are at play.

**\*For correspondence:**
hyliang@ustc.edu.cn (HL);
lmahadev@g.harvard.edu (LM)

**Competing interests:** The authors declare that no competing interests exist.

## Introduction

Microtubules (MTs) are polar tubular polymers formed by the self-assembly of the protein tubulin. MTs are ubiquitous in eukaryotic cells, where they are a major component of the cellular cytoskeleton, and participate in a number of essential cellular functions, such as cell migration, morphogenesis, transport within cells and cell division (*Desai, 1997*; *Gupta et al., 2015*; *Huber et al., 2013*; *Fletcher and Mullins, 2010*). They are also involved in regulating the shape and dynamics of axons, cilia and flagella (*Mimori-Kiyosue, 2011*).

The basic building blocks of MTs are tubulin heterodimers. These are formed by α-tubulin and β-tubulin, two structurally similar globular proteins with mass of about 55 kDa. αβ-tubulin dimers are arranged longitudinally into flexible tubulin filaments called protofilaments (PFs). A number (between 9 and 16, typically 13) of such PFs then assembles by lateral interactions to form the MT lattice (*Mandelkow et al., 1986*; *Chrétien and Wade, 1991*; *Mitchison, 1993*; *Tilney et al., 1973*). MT growth occurs by the addition of tubulin dimers mainly at the plus end, where β-tubulin is exposed. Upon hydrolysis of guanosine-tri-phosphate (GTP), tubulin subunits undergo a structural conversion that weakens lateral bonds, destabilises the subunit in the MT lattice and converts the relatively straight tubulin state into a state that is bound to guanosine-di-phosphate (GDP) and is characterised by an increased longitudinal curvature (*Wang and Nogales, 2005*; *Alushin et al., 2014*).

MTs are not static assemblies. They can repeatedly and stochastically vary their length by undergoing alternating phases of assembly and disassembly both in vivo and in vitro. This phenomenon is termed 'dynamic instability' and it is essential to a number of cellular functions, such as chromosome

separation, the remodelling of spatial organisation of the cytoskeleton during mitosis or the exploration of extracellular environment (*Mitchison and Kirschner, 1984*; *Gildersleeve et al., 1992*; *Cassimeris et al., 1988*; *Sammak and Borisy, 1988*). Understanding the factors that regulate MT dynamic instability is central to cell physiology and disease. Yet a detailed understanding of dynamic instability still remains elusive (*Aher and Akhmanova, 2018*; *Hemmat et al., 2018*). This difficulty originates in part from the fact that dynamic instability is the result of several mechanical and kinetic aspects operating at multiple time and length scales (*Figure 1*).

As such, dynamic instability of MTs has been the focus of extensive experimental and theoretical work (*Mandelkow et al., 1986*; *Chrétien and Wade, 1991*; *Mitchison, 1993*; *Tilney et al., 1973*; *Mitchison and Kirschner, 1984*; *Fygenson et al., 1994*; *Gildersleeve et al., 1992*; *Cassimeris et al., 1988*; *Sammak and Borisy, 1988*; *Hemmat et al., 2018*; *Aher and Akhmanova, 2018*; *Dogterom and Leibler, 1993*; *Brun et al., 2009*; *Antal et al., 2007*; *Mahadevan and*

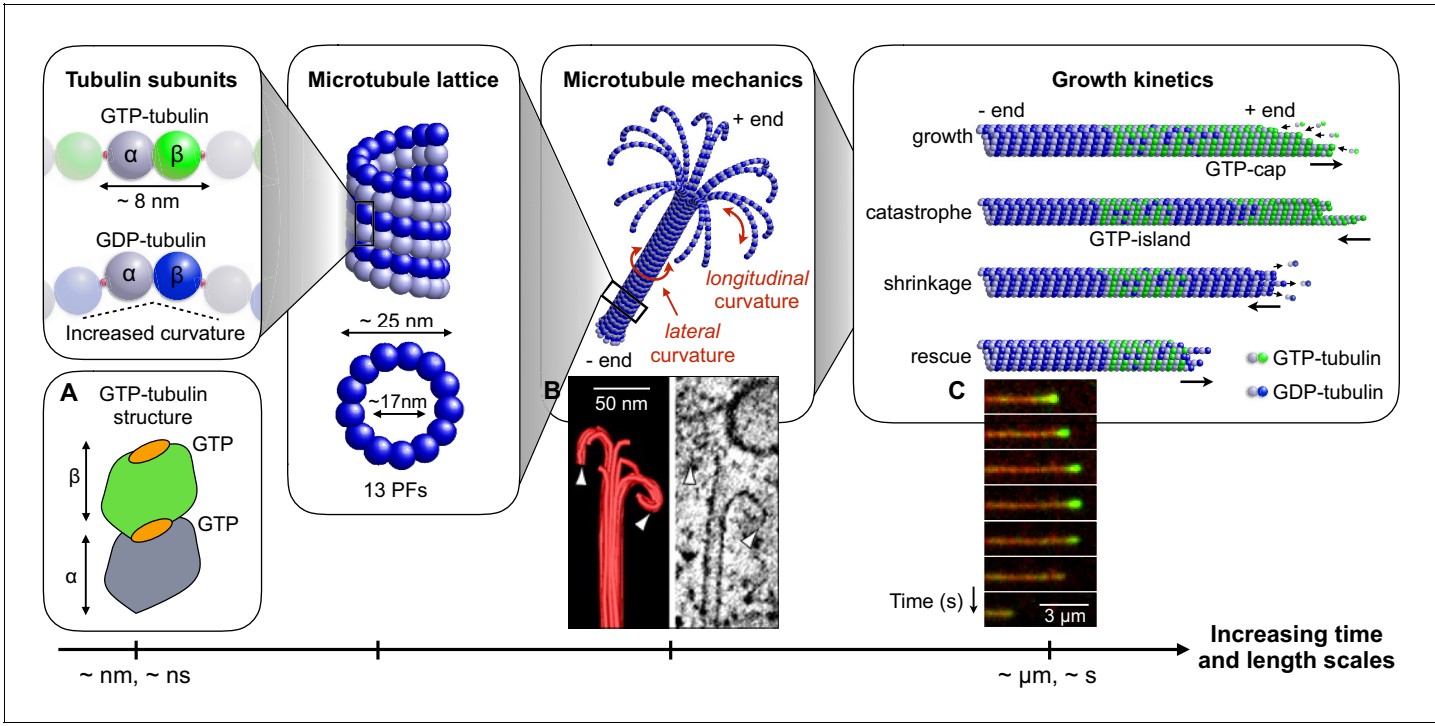

**Figure 1.** Connecting mechanical and kinetic aspects of MT dynamic instability across multiple time and length scales. MTs are composed of tubulin heterodimers formed of α-tubulin and β-tubulin. Both α-tubulin and β-tubulin are bound to GTP, but only the GTP that is bound to a β-tubulin is hydrolysable (*Nogales et al., 1998*). When the tubulin dimer is part of the MT lattice, GTP hydrolysis increases the spontaneous longitudinal curvature along the dimer axis. This causes GDP-tubulin dimers to be less tightly bound in the MT lattice (*Wang and Nogales, 2005*; *Alushin et al., 2014*). In the scheme, GTP-tubulin dimers are shown in green, while GDP-tubulin dimers are shown in blue; dark color indicates β-tubulin and light colour indicates α-tubulin. Tubulin dimers are connected head-to-tail into PFs. Typically, 13 such PFs align laterally to form the MT lattice, which is a long and hollow cylindrical shell with an outer diameter of approximately 25 nm and a thickness of about 5 nm (*Mandelkow et al., 1986*; *Chrétien and Wade, 1991*; *Mitchison, 1993*; *Tilney et al., 1973*). The mechanical stability/instability of the MT tube structure results from a competition between lateral and longitudinal curvatures. While MTs made of GTP-tubulin are relatively straight (about 5° per subunit), MTs made of GDP-tubulin tend to curve outward longitudinally at the plus end due to the longitudinal spontaneous curvature of GDP-tubulin dimers (about 12° per subunit) (*Chrétien et al., 1995*). Consequently, MTs consisting of GDP-tubulin (hydrolyzed MTs) tend to be mechanically less stable that their non-hydrolysed counterparts. The images show: (a) Schematic structure of an unhydrolyzed αβ-tubulin dimer with bound nucleotides highlighted in orange (*Alushin et al., 2014*). (b) EM image showing the characteristic shape of a depolymerising MT plus end, resembling a ram's horn (*VanBuren et al., 2005*). (c) Sequence of TIRF microscopy images of a MT end illustrating the switching between phases of polymerisation and depolymerisation during dynamic instability; catastrophe is associated with the loss of the GTP-caps (green fluorescence is from a protein that is believed to associate with the GTP cap). (a) is adapted from *Alushin et al. (2014)*; (b) is reproduced from *Austin et al. (2005)*. (c) is reproduced from *Duellberg et al. (2016)* under the Creative Commons Attribution License CC-BY-4.0 (https://creativecommons.org/licenses/by/4.0/).

© 2005 Company of Biologists Ltd. All rights reserved. Panel B is reproduced from *Austin et al. (2005)* with permission. It is not covered by the CC-BY 4.0 licence and further reproduction of this panel would need permission from the copyright holder.

*Mitchison, 2005*; *Nogales et al., 1998*; *Wang and Nogales, 2005*; *Alushin et al., 2014*; *Zovko et al., 2008*; *Hoog et al., 2011*; *Mandelkow et al., 1991*; *Chrétien et al., 1995*; *Austin et al., 2005*; *Gardner et al., 2011*; *Hunyadi and Jánosi, 2007*; *Rice et al., 2008*; *Kueh and Mitchison, 2009*; *Jánosi et al., 1998*; *Molodtsov et al., 2005*; *VanBuren et al., 2005*; *Zapperi and Mahadevan, 2011*; *Bertalan et al., 2014b*; *Wu et al., 2009*; *Duellberg et al., 2016*; *Seetapun et al., 2012*; *Cheng et al., 2012*; *Cheng and Stevens, 2014*; *Jain et al., 2015*; *Aparna et al., 2017*; *Zakharov et al., 2015*). Studies initially invoked a kinetic view capturing MT dynamic instability phenomenologically by different rates of polymerisation and depolymerisation depending on the state of the tubulin-phosphate complex (*Dogterom and Leibler, 1993*; *Brun et al., 2009*; *Antal et al., 2007*). Evidence of changes in the structural properties of MT subunits during hydrolysis later showed how lattice-bound tubulin dimers undergo a structural conformational change that increases their curvature (*Wang and Nogales, 2005*; *Alushin et al., 2014*; *Mahadevan and Mitchison, 2005*), suggesting that outward curving tips of hydrolysed MTs can cause such structures to be mechanically unstable (*Zovko et al., 2008*; *Hoog et al., 2011*; *Austin et al., 2005*; *Mandelkow et al., 1991*; *Chrétien et al., 1995*; *Gardner et al., 2011*; *Hunyadi and Jánosi, 2007*; *Rice et al., 2008*; *Kueh and Mitchison, 2009*). Several coarse-grained computer simulations have since then adopted this structural-mechanical view, considering MT elasticity explicitly, to understand different aspects of dynamic instability, including hydrolysis-driven mechanical deformations near the cap (*Jánosi et al., 1998*), force generation by shrinking microtubules (*Molodtsov et al., 2005*), or 3D sheet-like/blunt tips (*VanBuren et al., 2005*), or stochastic microtubule tip configurations and their relation to catastrophe (*Zakharov et al., 2015*). Few studies, however, have attempted to capture this mechanical view with the aim of emphasizing the qualitative features necessary for dynamic instability and providing phase diagrams that delineate the zones where dynamic instability is seen. Those that exist, for example (*Zapperi and Mahadevan, 2011*; *Bertalan et al., 2014b*), focus on the 1D limit, where MTs are modelled as adsorbed chains: the predictions from these models can be qualitatively different from those that correctly account for the 3D geometry of MTs (see 'Mechanical stability of 3D MTs in the presence of quenched disorder').

Here, we use a combination of theory and simulations to establish how the kinetics of polymerization and the mechanics associated with the 3D geometry of MTs act together across multiple scales to regulate dynamic instability. We also establish the role of disordered remnants of GDP-tubulin in determining the statistics of MT rescue. All together, our study provides a set of qualitative phase diagrams that delineate the regions of parameter space where dynamic instability is seen, consistent with previous observations while providing experimentally testable predictions.

## Methods

### Computational model

To complement the theory (see 'A phase diagram for mechanical stability of 3D MTs'), we developed a minimal coarse-grained computational model of MT mechanics and dynamics. To characterize the tubulin heterodimers, we use two patchy spheres linked together by a flexible hinge. We derive the patchy particles from coarse-grained representations of colloidal particles that are decorated by patches on their surface; the patches represent specific anisotropic interactions that promote binding with patches on other particles. In our model, interactions between dimers are described by patches carrying two types of interactions: longitudinal and lateral contacts (*Huisman et al., 2008*). Longitudinal contacts link dimers head-to-tail, arranging them into PFs. Lateral contacts connect parallel PFs to form the cylindrical shell of the MT. Each dimer has two longitudinal contacts and four lateral ones, corresponding to three patches per monomer (*Figure 2a*). Interactions between two patches on monomers $i$ and $j$ are described by the following potential (*Feng and Liang, 2012*):

$$V_{patchy}(r, \theta_i, \theta_j) = V_S(r, \theta_i, \theta_j) + V_B(r, \theta_i, \theta_j) + V_T(r, \theta_i, \theta_j). \tag{1}$$

We see that there are three distinct contributions associated with the stretching ($V_S$), bending ($V_B$) and twisting ($V_T$) modes, and we choose the following forms for these potentials:

$$V_S(r,\theta_i,\theta_j) = \begin{cases} \epsilon\left[\left(1 - e^{-a(r-r_0)}\right)^2 - 1\right] & r < r_m \\ \left[(1 - e^{\cdots} - \cdots\right]D_\theta(\theta_j) & r_m \le r < r_c \\ \cdots \ge r_c \end{cases} \tag{2}$$

$$V_B(r,\theta_i,\theta_j) = b(\theta_i^2 + \theta_j^2)D_r(r)D_\theta(\theta_i)D_\theta(\theta_j) \tag{3}$$

$$V_T(r,\theta_i,\theta_j) = c(\phi/r)^2 D_r(r)D_\theta(\theta_i)D_\theta(\theta_j) \tag{4}$$

where

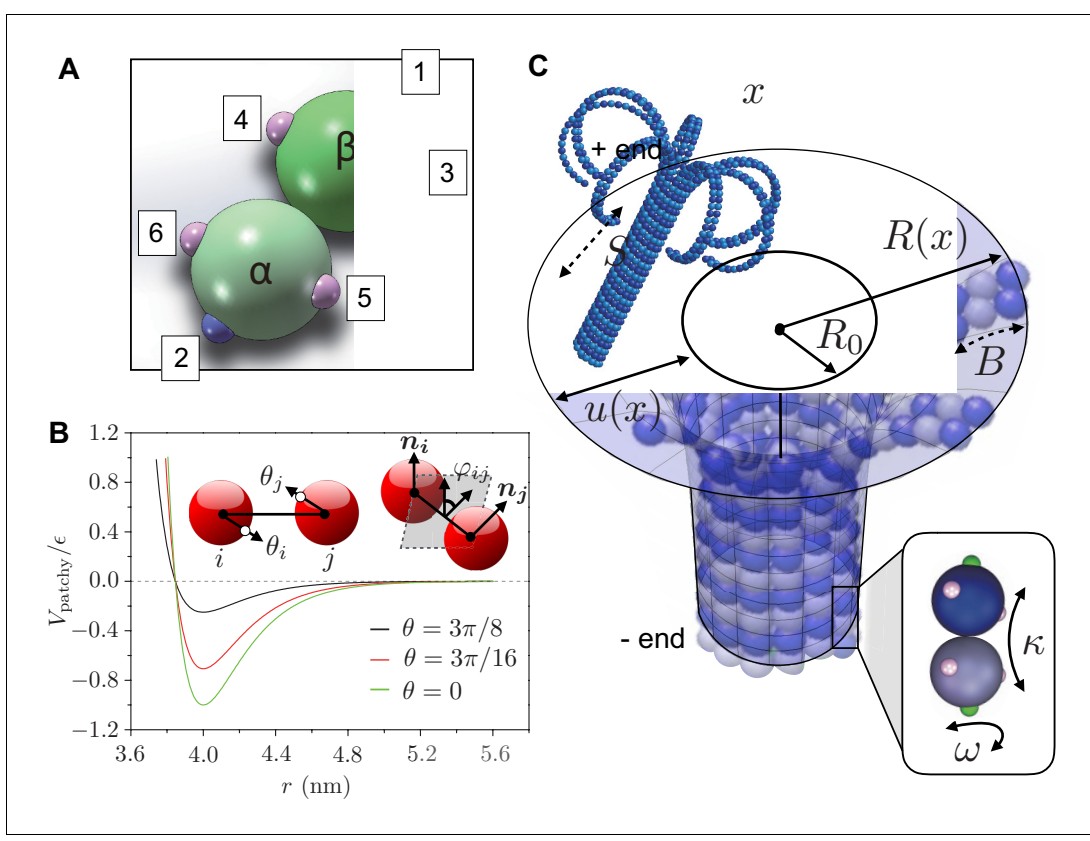

**Figure 2.** Definition of computer simulation and mechanical model. (a) The GTP-tubulin subunit ($\alpha\beta$-heterodimer) in our coarse-grained computer model. The blue patches (1 and 2) are longitudinal connecting points, while the pink patches (3 to 6) are lateral ones. This setup allows the dimer to curve inward along the MT longitudinal direction and outward laterally. The positions of the patches allow us to control the longitudinal and lateral curvatures of the subunit, giving a strong curvature (12°) to hydrolysed dimers and a weaker curvature (5°) to non-hydrolysed dimers. (b) The interaction potential $V_{\text{patchy}}(r;\theta_i,\theta_j)$ between two patchy particles as a function of their centre-to-centre distance $r$ for $\theta_i = 0$ and increasing $\theta_j = \theta$. Inset: the bending angles $\theta_i$ and $\theta_j$ are defined as the angles between the centre-to-patch line and the centre-to-centre line; the twisting angle $\varphi_{ij}$ is defined by the projection of the normals $n_i$ and $n_j$ on the plane perpendicular to the centre-to-centre line. (c) Our mechanical model describes a MT as a continuous, thin elastic sheet. This is parameterised as a surface of revolution obtained by rotating the function $R(x) = R_0 + u(x)$ along the MT long axis ($x$-axis), where $R_0$ is the natural radius of the MT and $u(x)$ describes the local deformation of the surface away from $R_0$. We approximate lateral interaction potentials between tubulin subunits in neighbouring PFs by a set of spring potentials (springs with stiffness $S$). Bending of the PFs in the MT causes the connecting springs to stretch. This competition between bending energy and stretching energy determines the mechanical stability of MTs. Inset: hydrolysed tubulin dimers curve naturally in two directions as described by the longitudinal and lateral spontaneous curvatures $\kappa$ and $\omega$, respectively.

$$D_r(r) = \begin{cases} 1 - \sin^4\left(\frac{\pi}{4}\frac{r-r_0}{r_d-r_0}\right) & r < r_d \\ 0 & r \geq r_d \end{cases} \tag{5}$$

$$D_\theta(\theta) = \begin{cases} 1 - \sin^4\left(\frac{\pi}{4}\frac{\theta}{\theta_d}\right) & \theta < \theta_d \\ 0 & \theta \geq \theta_d \end{cases} \tag{6}$$

Here, $r$ denotes the center-to-center distance between monomers, while the angles $\theta_i$ and $\theta_j$ describe the spatial directions of the patches (*Figure 2b*). The Morse potential term $V_S$, defined in *Equation 2*, describes the non-covalent interaction between patches, with $\epsilon$ being the depth of the potential well, $r_0$ the equilibrium distance between monomers, and $a$ is a parameter that controls the curvature of the potential well and, hence, determines the stretching modulus (see *Equation 7*). When $r < r_m$, where $r_m = r_0 - \log(2)/a$, the potential $V_S$ behaves as an isotropic repulsive interaction. In the range $r_m \leq r < r_c$, where $r_c = 5r_0$ is the cutoff for $V_S$ ($V_S$ is set to zero for $r \geq r_c$), $V_S$ is modified by multipliers $D_\theta(\theta_i)$ and $D_\theta(\theta_j)$. This yields an anisotropic attractive potential that exists only when the patches are aligned. Indeed, the multipliers $D_\theta(\theta_i)$ and $D_\theta(\theta_j)$, which are defined in *Equation 6*, weaken the attraction between the patches when these are not aligned (*Figure 2c*): $D_\theta$ reaches its maximum value when $\theta = 0$. The cutoff of $D_\theta$ is $\theta_d = \pi/3$ and limits the influence of the patches within particular range of spatial directions. The potential terms $V_B$ and $V_T$, defined in *Equations 3 and 4*, characterise bending and twisting deformations respectively. They are described as classical harmonic potentials with curvature $b$, respectively, $c$, and are modified by the multipliers $D_r(r)$, $D_\theta(\theta_i)$ and $D_\theta(\theta_j)$, which are defined in *Equations 5 and 6*. These multipliers limit the range of $V_B$ and $V_T$ to specific spatial locations and directions. The cutoff of $D_r$ is set as $r_d = 2.7r_0$, which is smaller than $r_c$ (the cutoff of $V_S$). This choice makes $V_B$ and $V_T$ shorter-range interactions compared to $V_S$.

The parameters in our coarse-grained computational model are linked to the mesoscopic mechanical properties of MTs (see section 'Mechanics') including the interfilament spring stiffness $S$, the filament bending stiffness $B$ and the filament torsional rigidity $K$ as

$$S = 2a^2 l\epsilon, \quad B = bl, \quad K = \frac{8c}{l}, \tag{7}$$

where $l$ is the length scale of tubulin dimers. $l$ takes different values depending on whether we are calculating longitudinal or lateral properties. In particular, we set $l = 2r_0 = 8$ nm when calculating longitudinal properties and set $l = r_0 = 4$ nm for lateral ones (see Appendix 1 and *Table 1* for further details on the computer simulation model and a summary of parameter choices). We choose the potential-well parameters in our simulations such that the resulting mesoscopic mechanical parameters *Equation 7* are consistent with the experimentally measured values of the mechanical properties of typical MTs (*Gittes et al., 1993*; *Mickey and Howard, 1995*; *Felgner et al., 1996*; *Tolomeo and Holley, 1997*; *de Pablo et al., 2003*; *Sept and MacKintosh (2010)*; *Deriu et al., 2010*). Simulations of this coarse-grained model of MTs were performed using Molecular Dynamics (MD), as described in Appendix 1.

## Results

### A phase diagram for mechanical stability of 3D MTs

Our analytical model to study the mechanical stability of 3D MTs is a function of the underlying parameters describing MT mechanics, MT growth kinetics and subunit hydrolysis, which we first consider in the deterministic limit (see 'A phase diagram for mechanical stability of 3D MTs'). This forms the basis for studying the mechanical stability of MTs when there is heterogeneity in the state of tubulin, that is it could be either GTP or GDP bound (see 'Mechanical stability of 3D MTs in the presence of quenched disorder'), and allows us to investigate the role of GTP-remnants (containing random fractions of non-hydrolysed subunits) on rescue (see 'Role of GTP-remnants in rescue').

The starting point of our mechanical model is that MTs exist as individual polymers with persistence lengths in the O(mm) range, which is much larger than the typical length of MTs (µm range) (*Fletcher and Mullins, 2010*; *Huber et al., 2013*). This observation suggests that MTs can be

**Table 1.** Values of parameters used in our coarse-grained simulations.

| Parameter | Value | Description |
|---|---|---|
| $r_0$ | 4 nm | Equilibrium distance between tubulin monomers |
| $\epsilon_0$ | $7.665 \times 10^{-20}$ J | Potential energy depth for longitudinal interactions |
| $\xi$ | 5° or 12° | Angle between α-tubulin and β-tubulin within a GTP- or GDP-dimer |
| $\chi_{chain}$ | 3° | See **Equations 24-29** |
| $\chi_{side1}$ | 103.45° or 90° | See **Equations 24-29** |
| $\chi_{side2}$ | 13.8° | See **Equations 24-29** |
| $l_{long}$ | $2r_0$ | Longitudinal size of tubulin dimers |
| $l_{lat}$ | $r_0$ | Lateral size of tubulin dimers |
| $a_{long}, a_{lat}$ | $20/r_0$ | Parameters that control longitudinal and lateral stretching stiffness |
| $b_{long}$ | $20\epsilon_0$ | Parameter that controls longitudinal bending stiffness |
| $b_{lat}$ | $10\epsilon_0$ | Parameter that controls lateral bending stiffness |
| $c_{long}$ | $500\epsilon_0 r_0^2$ | Parameter controls longitudinal twisting stiffness |
| $c_{lat}$ | $0.5\epsilon_0 r_0^2$ | Parameter that controls lateral twisting stiffness |
| $\epsilon_{long}$ | $\epsilon_0$ | Potential energy depth for longitudinal interactions |
| $\epsilon_{lat}$ | $0.0906\epsilon_0$ | Potential energy depth for lateral interactions |

considered to have a well-defined shape that is not affected significantly by thermal fluctuations. Moreover, previous studies indicate that lateral bonds between tubulin dimers are considerably weaker than longitudinal ones (**VanBuren et al., 2002**; **VanBuren et al., 2005**; **Molodtsov et al., 2005**). We model MTs as a set of adherent PFs that have bending stiffness $B$; we approximate lateral interactions between PFs by a series of spring potentials (springs of stiffness $S$) and assume that extending these springs beyond a critical displacement causes MTs to become mechanically unstable and break, potentially leading to dynamic instability.

## Kinetics

Previous studies (**Wang and Nogales, 2005**; **Alushin et al., 2014**) suggest that tubulin dimers that are part of the MT lattice have different mechanical properties depending on their hydrolysis state. In particular, upon hydrolysis the tubulin dimer undergoes a structural transformation from a relatively straight state to a state with finite curvature. In a 3D setting, the tubulin dimer can be curved both in the longitudinal direction and in the lateral direction. Let $\kappa(0)$ denote the longitudinal curvature of tubulin dimers in their GTP-state and let $\kappa(\infty)$ be the longitudinal curvature in the hydrolysed state. For simplicity, we assume that the hydrolysis reaction affects primarily the longitudinal curvature of the tubulin dimers, such that their curvature $\omega$ in the azimuthal direction can be considered to be constant. This assumption can be relaxed, see Appendix 2. Thus, as a result of GTP-hydrolysis, the longitudinal curvature of tubulin dimers, $\kappa$, changes with time, which we assume follows first order kinetics so that

$$\frac{d\kappa}{dt} = k_H(\kappa(\infty) - \kappa), \tag{8}$$

where $k_H$ is the rate of hydrolysis. While bound tubulin changes its structure, unbound (bound) tubulin can attach (detach) to (from) the free end of the MTs, which we also describe using a minimal first order kinetic law for the evolution of the length of the MT $n(t)$ (expressed in number of subunits) so that

$$\frac{dn}{dt} = k_+[m] - k_- = k_G. \tag{9}$$

Here $k_G$ is the net growth rate, $k_+$ is the elongation rate constant, $k_-$ is the dissociation rate constant and, for simplicity, we have assumed a constant subunit concentration $[m]$ in solution.

## Mechanics

In addition to MT growth and subunit hydrolysis, we also need to account for the elastic deformation of the MTs since the geometric state of the assembly is linked to its mechanical state. Individual PFs can bend, but are also constrained by inter-filament interactions, so that there are two contributions to elastic energy: (i) curvature energy associated with the bending of PFs, (ii) stretching energy of the springs connecting neighbouring PFs. We capture these energy contributions in a continuum picture that describes a MT as a thin elastic surface of revolution obtained by rotating the function $R(x) = R_0 + u(x)$ along the long MT axis (x-axis), where $R(x)$ is the local radius of MT and $R_0$ is the natural radius (*Figure 2c*). In the small gradient approximation, corresponding to $u' \ll 1$, the total elastic energy can be written as (see Appendix 2 for details):

$$\mathcal{E}_{tot} = \int_0^\infty \left( \frac{B}{2}[u'' - \kappa]^2 + \frac{\Sigma}{2}[u']^2 + \frac{S}{2}u^2 \right) dx, \tag{10}$$

where $\Sigma = B(1 + \omega R_0)^2/R_0^2$ and $' = \partial/\partial x$ denotes derivative with respect to $x$. The first term in *Equation 10* is the energy of MT that penalises deviations from its natural curvature $\kappa(t)$ which itself

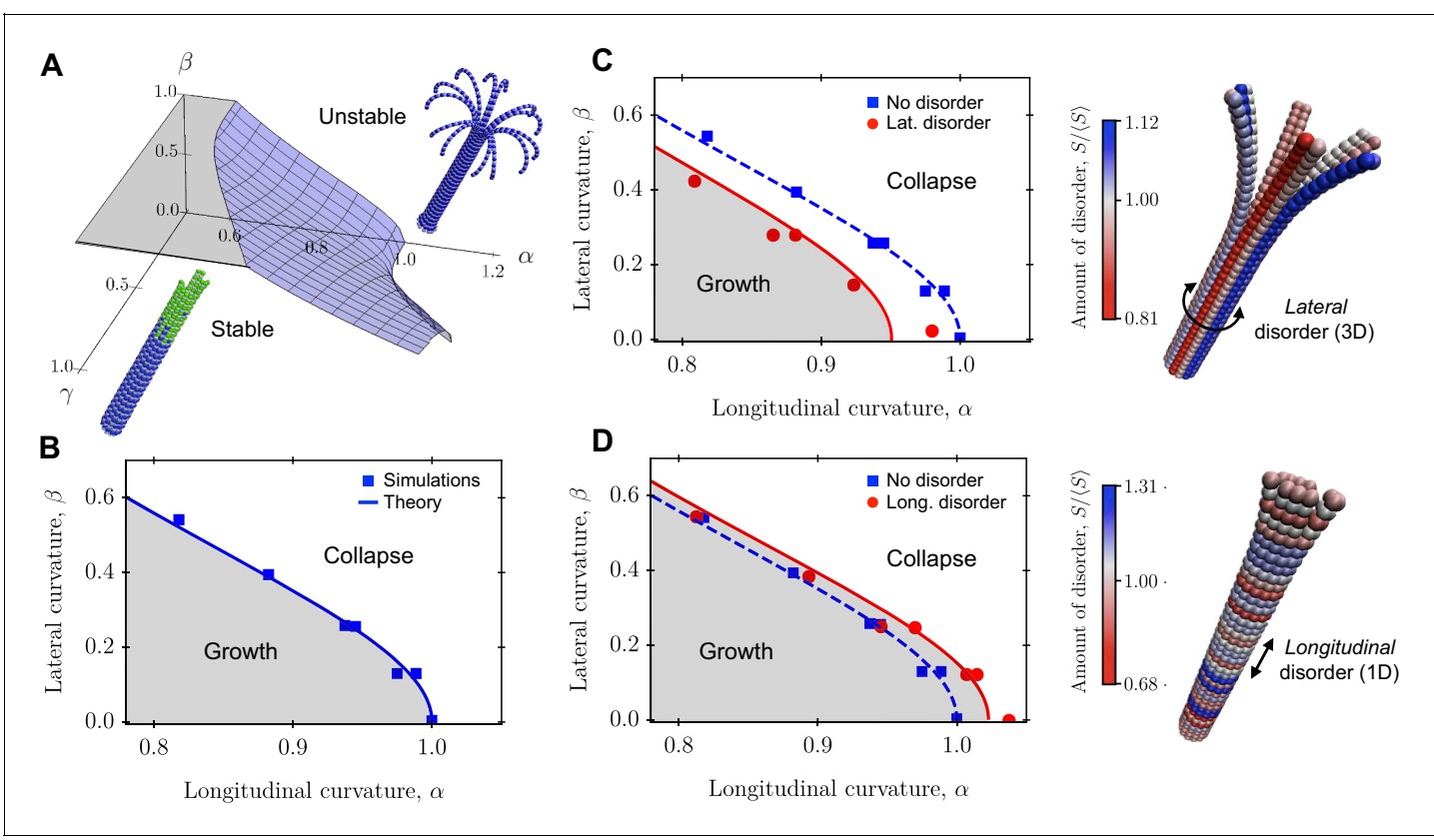

**Figure 3.** Role of lateral and longitudinal curvatures in mechanical stability of a 3D MT in the absence and presence of quenched disorder. (a) Schematic phase diagram for mechanical stability/instability with three axes (two mechanical axes and one kinetic axis): longitudinal curvature parameter $\alpha = (B\kappa^2/Su_c^2)^{1/2}$, lateral curvature parameter $\beta = (\Sigma u_c/B\kappa)^{1/2}$, and ratio of growth rate to hydrolysis rate $\gamma = \tau_\kappa/\tau_G$. (b) 2D phase diagram separating regions of mechanically stable and unstable MTs as a function of $\alpha$ and $\beta$. Data points are from computer simulations and the solid line is the prediction of *Equation 13*. (c)-(d) Implementation of lateral (c) and longitudinal (d) disorder at the level of the spring stiffness $S$ in our coarse-grained computer model. The data points are simulation results and the solid lines are fits to *Equation 16* and *Equation 17*. The data show that lateral disorder acts to destabilise MTs mechanically (c), while longitudinal disorder strengthens MTs (d). In both cases, disorder was generated using the distribution of spring constants in *Equation 14* with $\sigma = 0.02\langle S \rangle$, that is the disorder parameter was $k = 50$. The amount disorder of each tubulin subunit represents the average value of the relative stiffness $S/\langle S \rangle$ of its two lateral interactions. See *Videos 1–4* for movies illustrating the mechanical stability of a MT with and without quenched disorder (see *Videos 1–4*).

The online version of this article includes the following source data for figure 3:

**Source data 1.** This spreadsheet contains the data for *Figure 3B,C and D*.

reflects its state of hydrolysis. The second term is a surface energy term that penalises area increase due to the outward curving of the surface. The third term is the stretching energy of the springs. The minimum energy configuration results from a competition between bending energy, which favours a natural curved MT state, and elastic spring energy, which favours a straight cylindrical MT configuration. The overall shape of the axisymmetric tubule is then obtained by solving the Euler-Lagrange equation associated with *Equation 10* (see Appendix 2 for details):

$$Bu'''' - \Sigma u'' + Su = 0 \tag{11}$$

subject to the boundary conditions $u(\infty, t) = u'(\infty, t) = 0$ (fixed minus end), $u''(0, t) = \kappa$ and $u'''(0, t) = 0$ (free plus end), and is coupled to the kinetic *Equations 8 and 9*.

## Condition for mechanical stability

There are three natural dimensionless parameters (two mechanical parameters and one kinetic parameter) in our model that read:

$$\alpha = \left( \frac{B\kappa^2}{Su_c^2} \right)^{1/2}, \quad \beta = \left( \frac{\Sigma u_c}{B\kappa} \right)^{1/2}, \quad \gamma = \frac{k_G}{k_H}. \tag{12}$$

The first parameter $\alpha$ describes the effect of longitudinal curvature. The second parameter $\beta$ pertains to lateral curvature. Coupling these mechanical parameters to the kinetics of subunit hydrolysis and MT growth introduces an additional relevant dimensionless parameter $\gamma$, which is the ratio of the rate of hydrolysis of GTP-tubulin dimers to the net rate of addition of GTP-subunits to the MT plus end (see Appendix 2 for details).

These parameters serve as the basis for a phase diagram for the mechanical stability of a 3D MT (*Figure 3a*). Assuming that a mechanical instability arises when the elastic MT is deformed so that the radial displacement crosses a critical value ($u(0) > u_c$), we can solve *Equation 11* in terms of the maximal deformation $u(0)$ to yield a condition for when the MT is mechanically unstable. Rewriting this in terms of the scaled longitudinal curvature yields a critical value (see the Appendix 2 for details):

$$\alpha = \frac{\sqrt{1 + \frac{4\beta^2}{1 - e^{-n/\gamma}}} - 1}{2\beta^2}. \tag{13}$$

above which MTs are mechanically unstable (the transition curve in the $\alpha\beta$-plane in *Figure 3a*). This critical value depends on the lateral curvature parameter $\beta$, which is related to MT radius through $\beta \simeq 1/R_0$. In particular, the critical value for $\alpha$ is maximal ($\alpha = 1$) when $\beta = 0$, that is $R_0 \to \infty$. This situation corresponds to the limit of a one-dimensional MT (*Zapperi and Mahadevan, 2011*). The critical value for $\alpha$ then decreases with increasing lateral curvature $\beta$, that is decreasing MT radius. Overall, these results suggest that MTs with smaller radius are mechanically less stable than MTs with larger

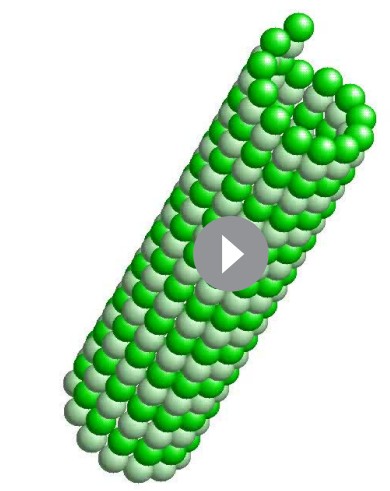

**Video 1.** MT growth and hydrolysis kinetics. This movie shows the interplay between MT growth kinetics (we focus here only on the addition of GTP-tubulin subunits at the plus end) and the subsequent hydrolysis of the incorporated subunits in the older parts of the MT. The interplay between growth and hydrolysis can be seen in the emergence of a growing front and a hydrolysis front that move with different speed. In this case, the hydrolysis front does not catch up with the growth front at the plus end, yielding a stabilising GTP-cap and a mechanically stable MT configuration for the entire duration of the simulation. The growth rate is $r_G = 10^{-4}$ steps$^{-1}$, while the probability of hydrolysis for each dimer is given by $p_H = e^{-r_H t}$, where $r_H = 10^{-10}$ steps$^{-1}$ is the rate of hydrolysis. Note the separation of timescales between MT growth and subunit hydrolysis ($r_G \gg r_H$). To keep the helical structure stable, we use the following parameters $b_{long} = 8\epsilon_0$, $b_{lat} = 4\epsilon_0$ and $c_{long} = 5\epsilon_0 r_0^2$, while the other mechanical parameters are the same as listed in *Table 1*.
https://elifesciences.org/articles/54077#video1

radius, consistent with the intuition that increasing azimuthal curvature increases the mechanical strain on the MT and thus makes it more likely to fracture. Furthermore, increasing the rate of MT growth over hydrolysis acts to stabilise MTs mechanically. In the mechanically stable phase, hydrolysis is slower than the addition of GTP-tubulin at the plus end, leaving a stabilising GTP-cap of size $n$ (see Appendix 2, *Video 1*). In the mechanically unstable phase, hydrolysis is faster than subunit addition at the plus end. Consequently, the 'hydrolysis front' takes over the 'growth front', which destabilises MTs. In this case, the PFs curve outward near the plus end, leading to a characteristic morphology of depolymerising MTs that resembles rams' horns (*VanBuren et al., 2005*) (see Appendix 2, *Video 2*). The transition curve separating these fast and slow hydrolysis regimes depends on both the longitudinal and lateral curvatures of the MT.

In *Figure 3b*, we show that crossing the transition curve given by *Equation 13* causes a switch from the mechanically stable phase into the mechanically unstable phase. This could result from variations in either $\alpha$, $\beta$ or $\gamma$. For MTs in the mechanically stable phase, catastrophic failure can still occur via thermal activation. In this regime, the rate of catastrophe follows Arrhenius' law $r_c \simeq \exp(\Delta E / k_B T)$, where $k_B T$ is the thermal energy and $\Delta E$ is the energy barrier given by $\Delta E \propto \alpha - 1 + \beta^2$ is a measure of the 'distance' from the transition curve *Equation 13* in the phase diagram of *Figure 3b*; the further away a MT is from this transition curve, the less likely it is to undergo catastrophe. The dependence of the rate of catastrophe on MT radius is through the parameter β, such that $\ln r_c \propto 1/R_0^2$. Thus, at constant temperature and at fixed values of the mechanical parameters, the rate of catastrophe increases with decreasing MT radius or, equivalently, decreasing PF number.

## Comparison with computer simulations

We used our computer simulations to test the prediction from *Equation 13* for how the critical value of $\alpha$ varies with β, which is a function of MT radius $R_0$ that is controlled by changing the number $N_f$ of PFs in the MT. The results (*Figure 3b*) show that the critical value for $\alpha$ is maximal when $\beta = 0$, and decreases with increasing β, in agreement with the theoretical prediction of *Equation 13* (solid line).

## Mechanical stability of 3D MTs in the presence of quenched disorder

Having considered the deterministic limit where growth kinetics, hydrolysis and longitudinal/lateral curvatures characterise the mechanical stability of a 3D MT, we now consider the role of randomness by including a random fraction of GTP-tubulin dimers in their lattice. We can model this situation by introducing quenched disorder in the state of the tubulin subunit. Quenched disorder describes the general situation when certain parameters in the system become random variables; disorder can be considered to be 'quenched' when the probability distribution of parameter values either does not vary with time or it varies with time slowly compared to some underlying fast dynamics, and thus cannot be described solely using equilibrium statistical mechanics. In the context of MTs, such a separation of timescales emerges very naturally when comparing fast polymerization/depolymerization kinetics and the comparatively slower GTP turn-over.

Since the primary mode of MT instability is due to the breaking of lateral bonds, a natural parameter for discussing the role of disorder is the spring stiffness $S$. The underlying motivation for this choice is that mechanical forces can influence the rates of chemical reactions. Indeed, mechanical work contributes to the free energy, which in turn determines the rates of a chemical reaction (*Howard, 2001*). In our context, GDP-

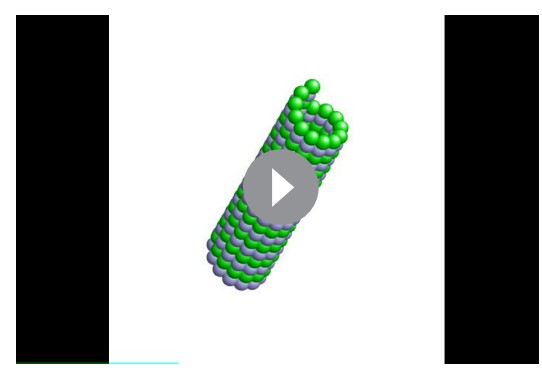

**Video 2.** MT catastrophe. This movie shows the interplay between MT growth and subunit hydrolysis. The rate of hydrolysis is $r_H = 5 \times 10^{-6}$ steps$^{-1}$ while the growth rate is decreasing from $r_G = 5 \times 10^{-3}$ steps$^{-1}$ to $r_G = 2 \times 10^{-6}$ steps$^{-1}$. Hydrolysis destroys the GTP-cap of MT and causes catastrophe. The mechanical parameters are the same as in **Video 1**.
https://elifesciences.org/articles/54077#video2

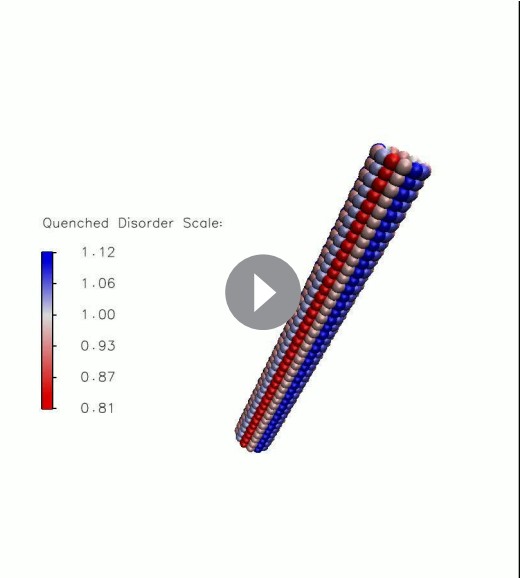

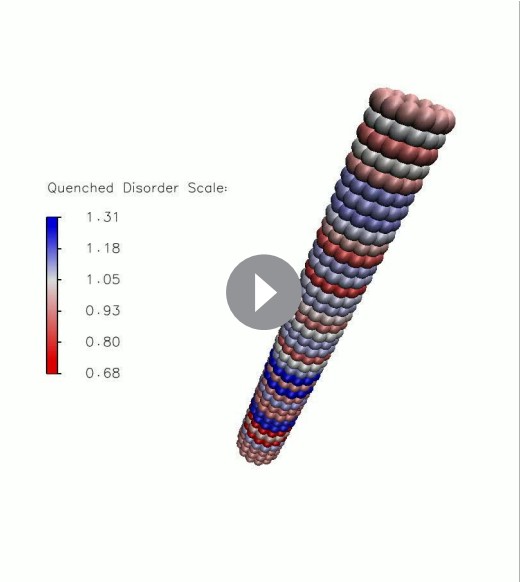

**Video 3.** Role of lateral quenched disorder in MT mechanical stability. This movie illustrates the stability of a MT in the presence of lateral disorder in the spring stiffness $S$, which describes the strength of lateral contacts between PFs. Note that lateral disorder causes the MT to be mechanically unstable along directions with weakest lateral bonds (the MT 'cut opens' along these weak directions). The color code indicates the local value of $S$ for each subunit, which is obtained by averaging over the relative stiffness of its two lateral bonds (drawn from the Gamma-distribution (14) with $k = 50$). The simulation parameters are the same as listed in **Table 1**, except that we set $\xi = 5.8°$ for all tubulin dimers.
https://elifesciences.org/articles/54077#video3

**Video 4.** Role of longitudinal quenched disorder in MT mechanical stability. This movie illustrates the stability of a MT in the presence of longitudinal disorder in the spring stiffness $S$. Note that the MT is in this case mechanically more stable than in the presence of lateral disorder. This is because longitudinal disorder leads to the presence of 'rings' of strong bonds that prevent the MT from peeling off completely. The simulation parameters are the same as in **Video 3**.
https://elifesciences.org/articles/54077#video4

tubulins within the MT lattice experience mechanical stresses due to the strong curvature. These mechanical stresses can shift the polymerisation-depolymerisation equilibrium and favour free monomers. This is consistent with the idea that GDP-tubulins are less tightly bound to MTs than GTP-tubulins (**Wang and Nogales, 2005**; **Alushin et al., 2014**), even if the chemical bonds are identical.

Thus, instead of having a well-defined spring constant $S$ throughout the MT, we consider a MT with varying $S$. Each lateral interaction is characterised in principle by a different spring constant $S$, which is drawn from a time–independent probability distribution $p(S)$ of spring constants. For convenience, we choose the Gamma-distribution

$$p(S; k, \langle S \rangle) = \frac{k^k S^{k-1} \exp(-kS/\langle S \rangle)}{\langle S \rangle^k \Gamma(k)}, \tag{14}$$

where $\Gamma(x)$ is the Gamma function, $\langle S \rangle$ is the average spring stiffness, and the parameter $1/k = \sigma/\langle S \rangle$, with $\sigma$ being the standard deviation of the distribution, is the coefficient of variation that describes the degree of disorder in the system. The choice of the Gamma-distribution admits a simple parameterisation in terms of the coefficient of variation that allows us to explore a range of different extreme value statistics. For instance, for $k = 1$ the Gamma distribution $p(S; 1, \langle S \rangle)$ yields the exponential distribution with intensity $\lambda = 1/\langle S \rangle$, while for $k \gg 1$ it yields a normal distribution with mean $\mu = k\langle S \rangle$ and variance $\sigma^2 = k\langle S \rangle^2$. We distinguish two limiting modes for disorder: lateral (**Figure 3c**) and longitudinal (**Figure 3d**). Any realisation of disorder can then be decomposed into a combination of these two limiting modes.

## Lateral disorder

In the presence of disorder in $S$, lateral interactions are characterised by variations in $S$ azimuthally but not longitudinally. In this case, whether the MT undergoes catastrophe depends on the breaking of the weakest lateral bond along the circumference of the MT, resulting in a MT that is 'cut-open' along the longitudinal direction (*Figure 3c* and Appendix 2, *Video 3*). This situation is fully analogous to what happens when pulling a one-dimensional chain by its ends: the chain will break as soon as its weakest link breaks. The mechanical stability of a MT with lateral disorder is thus equivalent to the mechanical stability of a MT with uniform spring stiffness $\langle S_{\min} \rangle$, where $\langle S_{\min} \rangle$ denotes the average value of the weakest spring stiffness along the MT circumference. This replacement maps the study of the mechanical stability of a MT with lateral disorder onto a problem of extreme value statistics (*Zapperi and Mahadevan, 2011*; *Bertalan et al., 2014b*): the determination of $\langle S_{\min} \rangle$ for a system of $N$ independent and identically distributed links with spring constants $S_1, \cdots, S_N$. In Appendix 2, we show that $\langle S_{\min} \rangle$ can be calculated from *Equation 14* using extreme-value statistics, yielding:

$$\frac{\langle S_{\min} \rangle}{\langle S \rangle} = \frac{1}{k} \, \Gamma\left(\frac{k+1}{k}\right) \left[\frac{\Gamma(k+1)}{N}\right]^{1/k}. \tag{15}$$

The condition for mechanical instability of a MT with lateral disorder in $S$ is thus obtained by replacing $\langle S \rangle$ by $\langle S_{\min} \rangle$ in *Equation 13*, yielding:

$$\alpha = \frac{\sqrt{1 + \frac{4\beta^2}{1 - e^{-n/\gamma}}} - 1}{2\beta^2} \sqrt{\frac{\langle S_{\min} \rangle}{\langle S \rangle}}. \tag{16}$$

Since $\langle S_{\min} \rangle < \langle S \rangle$, *Equation 16* predicts that the transition curve between mechanically stable and unstable MT regions shifts towards the instability region. Thus, lateral disorder weakens MTs.

## Longitudinal disorder

A different situation arises when quenched disorder is distributed longitudinally. Here, the strongest lateral bond determines MT stability. In fact, longitudinal disorder leads to the presence of 'rings' of particularly strong bonds that prevent the MT from depolymerising completely (*Figure 3d* and Appendix 2, *Video 4*). The mechanical stability of a MT with longitudinal disorder is thus equivalent to that of a MT with uniform spring stiffness $\langle S_{\max} \rangle$, where $\langle S_{\max} \rangle$ is the expected value of $S$ associated with the strongest lateral bond. Using extreme-value statistics, one finds (see Appendix 2) $\langle S_{\max} \rangle / \langle S \rangle = (\gamma_e + \log N)/k$, where $\gamma_e \approx 0.5772$ is the Euler-Mascheroni constant (*Taloni et al., 2018*; *Bertalan et al., 2014b*; *Zapperi and Mahadevan, 2011*). Hence, the curve separating mechanically stable and unstable regions in the presence of longitudinal disorder is *Zapperi and Mahadevan (2011)*:

$$\alpha = \frac{\sqrt{1 + \frac{4\beta^2}{1 - e^{-n/\gamma}}} - 1}{2\beta^2} \sqrt{\frac{\langle S_{\max} \rangle}{\langle S \rangle}}. \tag{17}$$

Since $\langle S_{\max} \rangle > \langle S \rangle$, longitudinal disorder in $S$ reinforces MTs.

It is important to note that longitudinal disorder is the only mode of disorder present in a 1D MT (*Zapperi and Mahadevan, 2011*). Lateral disorder is thus a defining feature of the 3D geometry of MTs. Our results thus reveal a fundamental role of MT dimensionality: while in a 1D setting quenched disorder stabilises MTs mechanically, in a 3D setting it can destabilise MTs.

## Comparison with computer simulations

We have tested the theoretical predictions of *Equation 16* and *Equation 17* using our coarse-grained simulations (*Figure 3c,d*). Quenched disorder was realised using *Equation 14* with disorder parameter $k = 50$. These simulations confirm that longitudinal quenched disorder increases the mechanical stability of MTs (*Figure 3c*), whereas the effect of quenched disorder in the lateral direction is to destabilise MTs mechanically (*Figure 3d*).

## Role of GTP-remnants in rescue

Using our theoretical model of mechanical stability of 3D MTs in the presence of quenched disorder, we are now in the position to investigate the role of remnants of GTP-tubulin in rescue. Rescue refers to the transition from depolymerisation to polymerisation during MT dynamic instability but is still poorly understood (*Brouhard, 2015*). Experiments indicate that GTP-tubulin addition at the plus end is not critical for rescue (*Gardner et al., 2013*; *Walker et al., 1988*), but that the presence of remnants of GTP-tubulin along the MT lattice in so-called 'GTP-islands' can lead to MT rescue (*Tropini et al., 2012*; *Dimitrov et al., 2008*; *Aumeier et al., 2016*; *Gardner et al., 2013*; *Vemu et al., 2018*). In particular, experiments in vivo have revealed a strong correlation between rescue probability and the presence of remnants of GTP-tubulin in older parts of the MTs, suggesting that these 'GTP-remnants' can function as rescue sites (*Dimitrov et al., 2008*). This view was further supported by the observation that the presence of a slowly hydrolyzable analogue of GTP bound to tubulin subunits contributes to MT rescue (*Tropini et al., 2012*. *Aumeier et al., 2016*) also demonstrated the possibility to generate GTP-islands along the MT lattice in a controlled manner by means of laser damaging and subsequent repair of the damaged site by incorporation of GTP-tubulin from solution (*Schaedel et al., 2015*): rescue occurred at laser-damaged sites in the presence of free GTP-tubulin (*Aumeier et al., 2016*). Separately, recent studies (*Vemu et al., 2018*; *Vemu et al., 2019*) reported of a damage-repair mechanism that stabilises MTs mediated by the enzymes spastin and katanin. Overall, these studies suggest that disordered GTP-islands in an otherwise structurally periodic lattice are involved in rescue regulation. Since these GTP-remnants are characterised by a random mixture of different states of tubulin, we ask if our framework might help to quantify these observations.

### Computer simulations

We first used our coarse-grained simulations to study the role of disordered GTP-remnants in MT rescue. We generated reinforcing islands by inserting, in the middle of a fully hydrolysed, depolymerising MT, a ring consisting of several layers of GTP-tubulin dimers (*Figure 4a*). We then observed whether the reinforcing GTP-islands were able rescue the depolymerising MTs as a function of two parameters: 1) the length of the GTP-island $N_{rf}$ (defined here as the number of layers in the island) and 2) the fraction $\phi$ of GTP-tubulin in the island. The results of these simulations (*Video 5*) are shown in *Figure 4b*. Note that the parameter $\phi$ controls the amount of disorder present in the island at the level of GTP-hydrolysis. This mimics both the scenario when rescue islands are formed because not all GTP-tubulin is able to hydrolyse, as suggested in *Dimitrov et al. (2008)*, or when rescue islands result from the incorporation of GTP-tubulin during the repair process of a damaged site, as suggested in *Aumeier et al. (2016)*. If disorder varies slowly over time compared to the characteristic timescale of polymerisation/depolymerisation, we can model slow changes of MT mechanical properties by making the relevant mechanical parameters explicit functions of time. For the parameters in our simulation, we find that when the reinforcing island is one layer long ($N_{rf} = 1$), the probability of rescue is close to zero, irrespective of the GTP-fraction in the reinforcing island. Interestingly, when $N_{rf} > 1$, we observe that rescue probability $p_{rescue}$ increases with $\phi$ in a highly nonlinear manner. Specifically, $p_{rescue}$ is either close to zero or close to one for most values of $\phi$, with a sharp increase in the transition region.

### Percolation model of rescue

To qualitatively understand the observed nonlinear behaviour of rescue probability with GTP-fraction in the reinforcing island $\phi$, we propose a site percolation model of rescue (*Figure 4c*). Site percolation is concerned with the following question: given a random graph, in which each site is (independently) occupied with probability $q$ or empty with probability $1 - q$, what is the probability that a connected path of occupied sites exists between the boundaries of the graph? In our percolation model of rescue, each site of the reinforcing island is occupied by a GTP-tubulin dimer with probability $\phi$, while it is occupied by a GDP-tubulin dimer with probability $1 - \phi$. In Sec. 'Role of quenched disorder in mechanical stability of MTs', we have shown that the presence of randomly distributed weak lateral bonds (mediated by GDP-tubulin) can destabilise MTs mechanically when disorder is longitudinal. As such, a MT will be mechanically unstable when a connecting path of GDP-tubulins runs longitudinally through the reinforcing island (*Figure 4c*). The question of whether a reinforcing

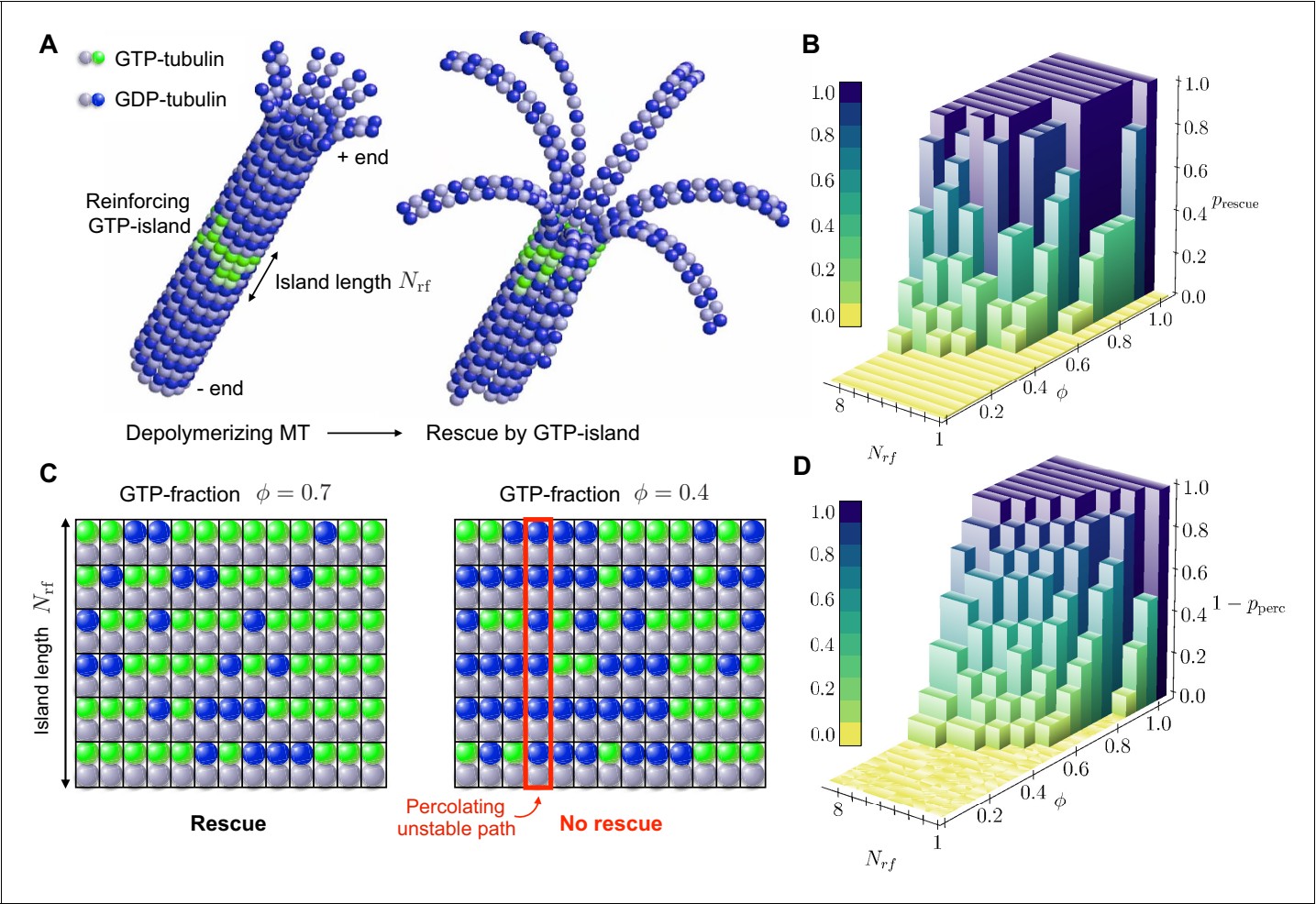

**Figure 4.** Role of disordered GTP-islands in rescue. (a) Schematics showing the rescue of a depolymerising MT reinforced by a disordered GTP-tubulin island. In this example, the length of the reinforcing island is $N_{\rm rf} = 3$ and the GTP-island consists of 20 GTP-tubulin dimers, which corresponds to a GTP-fraction of $\phi = 20/(3 \times 13) \simeq 0.51$ (see **Video 5**). (b) Simulated rescue probability by GTP-islands as a function of GTP-fraction $\phi$ and reinforcing island length $N_{\rm rf}$. The rescue probability $p_{rescue}$ was estimated by repeating simulations six to ten times for each pair $\phi$ and $N_{\rm rf}$. (c) Percolation model for the role of disordered reinforcing GTP-islands on MT rescue. (d) Simulated site percolation on a square lattice with dimensions $N_{\rm rf} \times 13$ as a function of GTP-fraction $\phi$ and island length $N_{\rm rf}$. The probability of percolation was obtained by averaging over $10^3$ realisations for each pair $\phi$ and $N_{\rm rf}$.
The online version of this article includes the following source data for figure 4:

**Source data 1.** This spreadsheet contains the data for **Figure 4B**.
**Source data 2.** This spreadsheet contains the data for **Figure 4D**.

island with GTP-fraction $\phi$ is able to rescue a depolymerising MT is thus analogous to site percolation with $q = \phi$. The rescue probability $p_{rescue}$ thus relates to $1 - p_{\rm perc}$, where $p_{\rm perc}$ is the probability of percolation of a longitudinal path of GDP-tubulin subunits through the length of the reinforcing island. **Figure 4d** shows that the results of site percolation on a square lattice of dimensions $13 \times N_{\rm rf}$ with varying $\phi$ are in qualitative agreement with simulated rescue probabilities (**Figure 4b**).

## Discussion

Our multi-scale approach to dynamic instability incorporates the mechanics and 3D geometry of MTs, the kinetics of tubulin addition and GTP-hydrolysis, and quenched disorder in the state of the tubulin subunit. Our results provide a series of phase diagrams for the presence of dynamic instability, revealing the dimensionless mechanical and kinetic parameters controlling the problem. Compared to previous analytic studies of dynamic instability, our results reveal the key role of the 3D

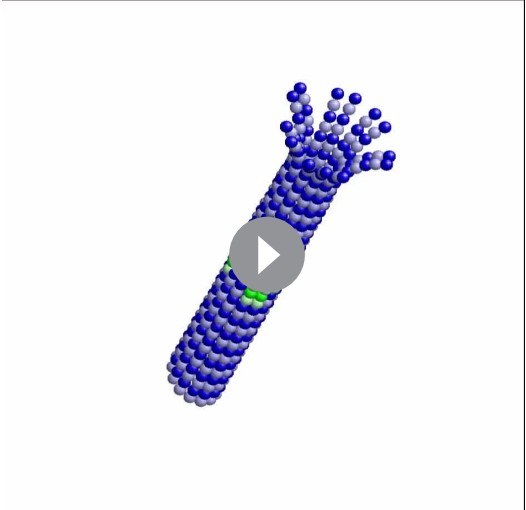

**Video 5.** MT rescue by disordered GTP-island. This movie illustrates the successful rescue of a depolymerising MT by a disordered GTP-island placed along the MT lattice. The mechanical parameters in this case are the same as in **Video 1**.
https://elifesciences.org/articles/54077#video5

geometry of MTs. In particular, we find that the mechanical stability of a MT is strongly affected by its radius (*Mahadevan and Mitchison, 2005*); a MT with a smaller radius is mechanically less stable than an identical MT with a larger radius.

Since MT radius has been shown to vary because the number $N_f$ of PFs composing MTs typically ranges between 9 to 16 (*Chrétien et al., 1992*; *Chalfie and Thomson, 1982*; *Chaaban and Brouhard, 2017*; *Cueva et al., 2012*; *Díaz et al., 1998*), our study suggests quantitative experimental tests via measurements of catastrophe rates $r_c$ as a function of PF number $N_f$ to verify the theoretical prediction for the rate of catastrophe $\log(r_c) \propto 1/N_f^2$. Another key prediction from our study is that the rescuing power of GTP-islands displays a sharp drop at intermediate values of GTP-fraction. In particular, the percolation model predicts that there is a critical point for the GTP-fraction, $\phi = \phi_c$, below which reinforcing islands lose their ability to rescue MT disassembly. The numerical value of the threshold $\phi_c$ depends on the thickness of the reinforcing island as well as on the MT lattice structure. This critical GTP-fraction could be determined experimentally and compared to theory by using non-hydrolyzable analogs of tubulin (*Tropini et al., 2012*), to control the amount of disorder at the level of the state of tubulin subunit in the island, in combination with super-resolution microscopy (*Huang et al., 2009*) to establish island length. Finally, we note that our assumption of the form of the quenched disorder in terms of the Gamma distribution is just that - an assumption. Further experimental work will be required to solve the inverse problem of estimating the average GTP-fraction in GTP remnants from rescue probabilities determined experimentally (*Dimitrov et al., 2008*; *Aumeier et al., 2016*; *Vemu et al., 2018*) to see if we might determine both the form of the disorder and the intrinsic parameters characterising it, thus allowing future research to address the question of how to control dynamic instability.

## Acknowledgements

We acknowledge financial support from the Swiss National Science foundation (TCTM), the National Natural Science Foundation of China (11272303,11072230) (SF, LH).

## Additional information

### Funding

| Funder | Grant reference number | Author |
|---|---|---|
| Swiss National Science Foundation | | Thomas CT Michaels |
| National Natural Science Foundation of China | 11272303 | Shuo Feng Haiyi Liang |
| National Natural Science Foundation of China | 11072230 | Shuo Feng Haiyi Liang |

The funders had no role in study design, data collection and interpretation, or the decision to submit the work for publication.

## Author contributions
Thomas CT Michaels, Conceptualization, Formal analysis, Investigation, Methodology, Writing - original draft, Writing - review and editing; Shuo Feng, Software, Validation, Visualization, Methodology; Haiyi Liang, Software, Supervision, Validation, Investigation, Methodology; L Mahadevan, Conceptualization, Formal analysis, Supervision, Funding acquisition, Validation, Investigation, Writing - original draft, Project administration, Writing - review and editing

## Author ORCIDs
Thomas CT Michaels (iD) https://orcid.org/0000-0001-6931-5041
Shuo Feng (iD) http://orcid.org/0000-0003-0530-8381
Haiyi Liang (iD) https://orcid.org/0000-0001-7458-8036
L Mahadevan (iD) https://orcid.org/0000-0002-5114-0519

## Decision letter and Author response
Decision letter https://doi.org/10.7554/eLife.54077.sa1
Author response https://doi.org/10.7554/eLife.54077.sa2

## Additional files
### Supplementary files
• Transparent reporting form

### Data availability
All data generated or analysed during this study are included in the manuscript and supporting files.

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

## Appendix 1

### Computer model

We assign a local coordinate system to each dimer to describe its overall rotation, as well as the relative position of the patches within the dimer. To describe the whole dimer, we need three basis vectors $t, s$ and $n$ is directed along the long dimer axis and points from the $\alpha$-tubulin to the $\beta$-tubulin. $s$ and $n$ lie in the plane perpendicular to $t$. This coordinate system is particularly convenient for describing the rotation of a dimer as a whole. However, it cannot capture the relative rotation of the $\alpha$- and $\beta$-tubulin monomers in the dimer. This inner rotation can only be along the axis $n$, and thus two patches coordinate systems (one for each tubulin in the dimer) can be obtained by rotating the local coordinate system around $n$ as:

$$t_\beta = t \cos(\xi/2) - s \sin(\xi/2), \tag{18}$$

$$s_\beta = t \sin(\xi/2) + s \cos(\xi/2), \tag{19}$$

$$n_\beta = n, \tag{20}$$

$$t_\alpha = t \cos(\xi/2) + s \sin(\xi/2), \tag{21}$$

$$s_\alpha = t \cos(\xi/2) + s \sin(\xi/2), \tag{22}$$

$$n_\alpha = n, \tag{23}$$

where $\xi$ is the angle between the vectors $t_\alpha$ and $t_\beta$ (the axes of the two monomers), as shown in **Appendix 1—figure 1a**.

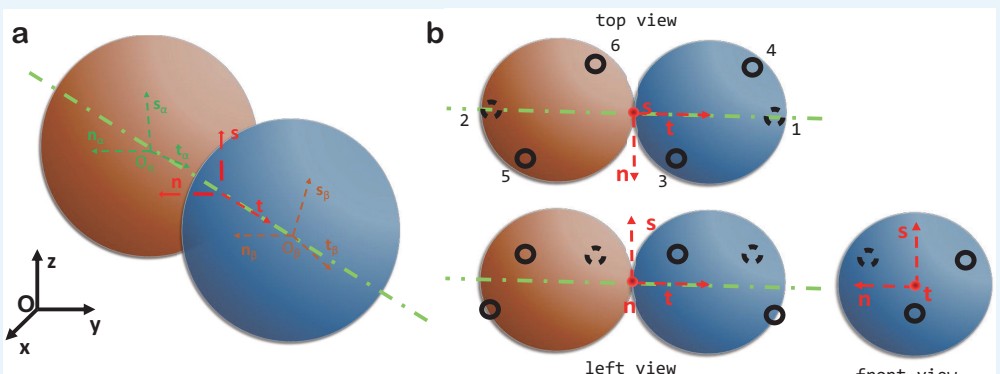

**Appendix 1—figure 1.** Computer model. (**a**) The local coordinate systems of a dimer. (**b**) Top, left and front views of two spheres forming a dimer and the six patchy points.

There are two types of connections between dimers: longitudinal and lateral contacts. Longitudinal contacts link dimers head to tail and arrange them into PFs. Lateral contacts connect parallel PFs arranging them into a cylindrical MT shell. Each dimer has two longitudinal contacts and four lateral ones, that is there are six patchy points on each dimer. These are numbered as in **Appendix 1—figure 1b** or **Figure 2a** of the main text. The coordinates of the six patchy points are described by the following equations:

$$R_1 = t_\beta \cos(\chi_{chain}) - s_\beta \sin(\chi_{chain}), \tag{24}$$

$$R_2 = t_\alpha \cos(\chi_{chain}) - s_\alpha \sin(\chi_{chain}), \tag{25}$$

$$\boldsymbol{R}_3 = (\boldsymbol{t}_\beta \cos(\chi_{side1}) - \boldsymbol{n}_\beta \sin(\chi_{side1})) \cos(\chi_{side2}) + \boldsymbol{s}_\beta \sin(\chi_{side2}), \tag{26}$$

$$\boldsymbol{R}_4 = (-\boldsymbol{t}_\beta \cos(\chi_{side1}) + \boldsymbol{n}_\beta \sin(\chi_{side1})) \cos(\chi_{side2}) + \boldsymbol{s}_\beta \sin(\chi_{side2}), \tag{27}$$

$$\boldsymbol{R}_5 = (\boldsymbol{t}_\alpha \cos(\chi_{side1}) - \boldsymbol{n}_\alpha \sin(\chi_{side1})) \cos(\chi_{side2}) + \boldsymbol{s}_\alpha \sin(\chi_{side2}), \tag{28}$$

$$\boldsymbol{R}_6 = (-\boldsymbol{t}_\alpha \cos(\chi_{side1}) + \boldsymbol{n}_\alpha \sin(\chi_{side1})) \cos(\chi_{side2}) + \boldsymbol{s}_\alpha \sin(\chi_{side2}). \tag{29}$$

The values of the angles $\chi_{chain}, \chi_{side1}, \chi_{side2}$ are given in *Table 1*.

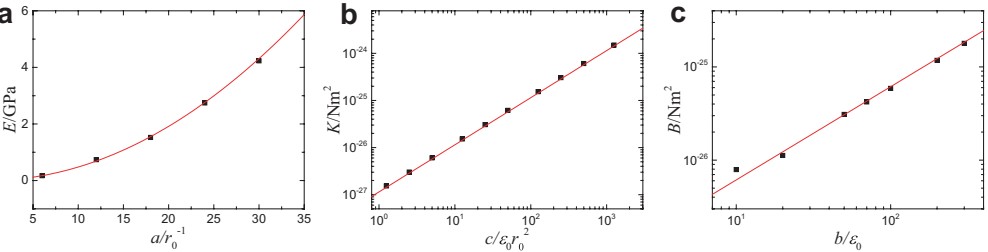

**Appendix 1—figure 2.** Mesoscopic mechanical properties of MTs as functions of the parameters in the coarse-grained simulations model. (**a**) Young's modulus $E$ vs $a$, (**b**) torsional rigidity $K$ vs $c$ and (**c**) bending stiffness $B$ vs $b$. Solid lines correspond to the theoretical predictions (see *Equation 7* in Section 'Computational model' of main text), and are in good agreement with the computer simulations.

## Appendix 2

### Theoretical model

Here we provide additional mathematical details pertaining to the theoretical model discussed in the main text, in terms of the elastostatics of the structure and kinetics of polymerisation/depolymerisation and hydrolysis. Finally, we study this model in the presence of quenched disorder.

### Mechanical stability of 3D MTs (statics)

Our model considers two contributions to the total elastic energy of MTs: 1) a *curvature energy* term describing the bending of PFs in the MT and 2) the *stretching energy* of the springs that connect neighboring PFs. We now consider these two energy contributions in detail.

#### Curvature energy

We describe the curvature energy term by means of the *Helfrich (1973)*:

$$\mathcal{E}_{\text{curvature}} = \int dA \left[ \frac{k_1}{2} (2H)^2 + k_2 K \right], \tag{30}$$

where $k_1$ and $k_2$ are bending rigidities, $H = (\kappa_1 + \kappa_2)/2$ is the mean curvature, $K = \kappa_1 \kappa_2$ is the Gaussian curvature, $\kappa_1$ and $\kappa_2$ are the principal curvatures, and $dA$ is the surface area element. We describe the MT as a continuous elastic sheet modelled as a solid of rotation obtained by rotating the function $R(x)$ along the $x$-axis, which describes the microtubule growth axis (*Appendix 2—figure 1a*). We assume that the MT possesses a natural radius $R_0$ (when flat) and, for convenience, we write $R(x) = R_0 + u(x)$, where $u(x)$ describes the deviation of local radius $R(x)$ from $R_0$. The MT surface is thus parameterised as $(x, R(x) \cos \varphi, R(x) \sin \varphi)$ where $x \in [0, \infty)$ and $\varphi \in [0, 2\pi)$. In this parametrisation, $x = 0$ corresponds to the MT plus end, while $x = \infty$ corresponds to the minus end. We then use the following expressions for the mean and Gaussian curvatures of a surface of revolution *Gray, 1997*:

$$H(x) = \frac{R(x) R''(x) - (1 + R'(x)^2)}{2R(x)[1 + R'(x)^2]^{3/2}}, \quad K(x) = \frac{-R''(x)}{R(x)[1 + R'(x)^2]^2}, \tag{31}$$

where $' = d/dx$ denotes derivative with respect to $x$, and the surface area element is $dA = 2\pi R(x) \sqrt{1 + R'(x)^2} dx$. From these expressions, we note that the Gaussian curvature term, $K(x) dA = -R''(x)/[1 + R'(x)^2]^{3/2} dx$, is fully integrable, that is it gives rise to boundary terms only. We thus focus on the mean curvature term only. To this end, we consider a small gradient approximation, which corresponds to $u'(x) \ll 1$. We then write $R(x) = R_0 + u(x)$ in *Equation 31* and after keeping only the leading order terms, we find

$$(2H)^2 dA \simeq 2\pi \left( R_0 u''(x)^2 + \frac{u'(x)^2}{2R_0} \right) dx + \mathcal{R}, \tag{32}$$

where $\mathcal{R}$ stands for higher order terms in $u'(x)$ or for boundary terms (i.e. terms that are fully integrable and thus, after integration, simply shift the curvature energy by a constant value).

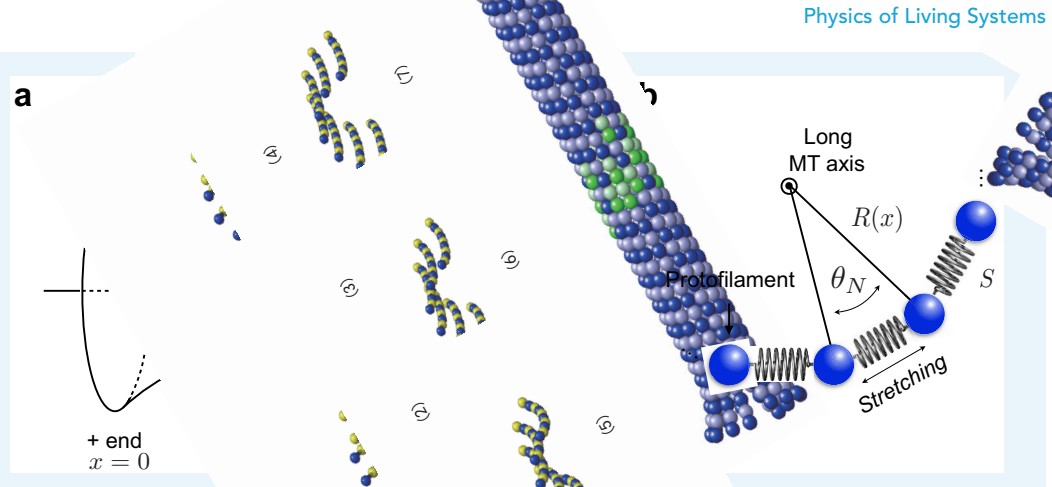

**Appendix 2—figure 1.** Continuum elastic shell model of microtubule. (**a**) To describe the curvature energy, MTs are modelled as thin elastic surfaces of revolution obtained by rotating the function $R(x) = R_0 + u(x)$ along the MT long axis ($x$ axis). $R_0$ is the natural radius of the MT. (**b**) Neighbouring PFs within the MT are connected by $N_f$ Hookean springs with stiffness $k_s$.

Our calculation so far has accounted for bending energy relative to a flat cylindrical MT. To account for the natural longitudinal and lateral curvatures of subunits ($\kappa$ and $\omega$, respectively), we extend the Helfrich Hamiltonian as follows:

$$\mathcal{E}_{\text{curvature}} = \int dA \left[ \frac{k_1}{2} (2H - 2H_0)^2 + k_2 K \right], \tag{33}$$

where $H_0 = (\kappa + \omega)/2$ is the natural (or spontaneous) mean curvature. In the small gradient approximation ($u' \ll 1$), the leading contribution to the curvature energy in the presence of natural curvatures $\kappa$ and $\omega$ is found to be

$$(2H - 2H_0)^2 dA \simeq 2\pi \left( R_0 [u''(x) - \kappa]^2 + \frac{u'(x)^2}{2} \frac{(1 + \omega R_0)^2}{R_0} \right) dx + \mathcal{R}. \tag{34}$$

Thus, the curvature energy is

$$\mathcal{E}_{\text{curvature}} \simeq \int_0^\infty \left( \frac{B}{2} [u''(x) - \kappa]^2 + \frac{\Sigma}{2} u'(x)^2 \right) dx, \tag{35}$$

where $B = 2\pi R_0 k_1$ and $\Sigma = B(1 + \omega R_0)^2 / R_0^2$.

## Elastic spring energy

The second contribution to the total elastic energy of MTs is due to the stretching energy of the springs connecting neighbouring PFs. To construct this energy contribution, we model the cross section of the MT as a set of PFs connected by $N_f$ harmonic springs with stiffness $k_s$ (**Appendix 2—figure 1b**). The stretching energy is proportional to the square of the deviations of the distance $d$ between PFs from their equilibrium position $d_0$. By symmetry in the $\varphi$ direction, springs are stretched by $d(x) \simeq \theta_N R(x) = 2\pi R(x)/N_f$ ($\theta_N$ is defined in **Appendix 2—figure 1b**). The rest length of springs is $d_0 = \theta_N R_0 = 2\pi R_0/N_f$. Thus, the stretching energy per unit length of the system of $N_f$ springs can be estimated as:

$$N_f \times \frac{k_s}{2} [d(x) - d_0]^2 \simeq \frac{N_f k_s}{2} \left( \frac{2\pi}{N_f} \right)^2 [R(x) - R_0]^2. \tag{36}$$

Thus, writing $R(x) = R_0 + u(x)$, the stretching energy is found to be:

$$\mathcal{E}_{\mathrm{stretch}} \simeq \frac{S}{2} \int_0^\infty u(x)^2 dx, \tag{37}$$

where $S = 4\pi^2 k_s / N_f$.

## Total elastic energy

In summary, the total elastic energy of a 3D MT can be written in terms of $u(x)$ by combining the contributions from curvature energy and spring energy:

$$\mathcal{E}_{\mathrm{tot}} = \mathcal{E}_{\mathrm{curvature}} + \mathcal{E}_{\mathrm{stretch}} \simeq \int_0^\infty \left( \frac{B}{2} [u''(x) - \kappa]^2 + \frac{\Sigma}{2} u'(x)^2 + \frac{S}{2} u(x)^2 \right) dx. \tag{38}$$

The first term describes the longitudinal bending energy of the MT away from the natural curvature $\kappa$. The second term comes from the lateral curvature. At leading order, this term is a surface energy term that penalises the increase in surface area when the MT curves out; the parameter $\Sigma$ plays the role of a surface tension. Finally, the third term is the stretching energy of the springs, which gives rise to an energy density contribution proportional to $u(x)^2$.

## Euler-Lagrange equation and minimum energy configuration

Having defined the elastic energy functional for the problem, $\mathcal{E}_{tot}[u] = \int_0^\infty \mathcal{H}(u, u', u'') dx$ (**Equation 38**), we now consider the associated Euler-Lagrange equation that describes the minimal energy configuration of MTs:

$$\frac{\delta \mathcal{E}_{tot}[u]}{\delta u} = \frac{\partial \mathcal{H}}{\partial u} - \frac{d}{dx}\left(\frac{\partial \mathcal{H}}{\partial u'}\right) + \frac{d^2}{dx^2}\left(\frac{\partial \mathcal{H}}{\partial u''}\right) = 0. \tag{39}$$

Using **Equation 38**, **Equation 39** is found to be:

$$Bu'''' - \Sigma u'' + Su = 0. \tag{40}$$

Subject to the following boundary conditions

$$u(\infty) = u'(\infty) = 0 \ (\mathrm{MT\,flat\,at}\,x = \infty, \mathrm{i.e.\,at\,MT\,minus\,end}), \tag{41}$$

$$u''(0) = \kappa \ (\mathrm{natural\,curvature\,at}\,x = 0, \mathrm{i.e.\,at\,MT\,plus\,end}), \tag{42}$$

$$u'''(0) = 0 \ (\mathrm{no\,shear\,force\,in\,the\,MT}) \tag{43}$$

the analytical solution to **Equation 40** reads:

$$u(x) = \frac{\kappa \lambda_2}{\lambda_1^2(\lambda_2 - \lambda_1)} e^{-\lambda_1 x} + \frac{\kappa \lambda_1}{\lambda_2^2(\lambda_1 - \lambda_2)} e^{-\lambda_2 x}, \tag{44}$$

where

$$\lambda_{1,2} = \frac{\sqrt{\frac{1}{\ell_\alpha^2} \pm \sqrt{\frac{1}{\ell_\alpha^4} - \frac{4}{\ell_\beta^4}}}}{\sqrt{2}}, \tag{45}$$

and the two relevant length scales in the problem are

$$\ell_\alpha = \left(\frac{B}{\Sigma}\right)^{1/2}, \quad \ell_\beta = \left(\frac{B}{S}\right)^{1/4}. \tag{46}$$

The first length scale is associated with the lateral curvature of the MT. In fact, from $\Sigma = B(1 + \omega R_0)^2 / R_0^2$ it follows $\ell_\alpha = (B/\Sigma)^{1/2} \simeq R_0$.

## Condition for mechanical stability/instability of 3D MTs

To study the mechanical stability/instability of MTs as a function of its mechanical parameters, we assume that the springs connecting PFs break if their extension exceeds a critical value $u_c$. From **Equation 44** we thus obtain the following condition for mechanical instability:

$$u(0) = \kappa \ell_\beta^2 \left( 1 + \frac{\ell_\beta^2}{\ell_\alpha^2} \right) > u_c. \tag{47}$$

To rewrite this in a more transparent way in terms of the dimensionless parameters, we first define the scaled longitudinal and lateral curvatures:

$$\alpha = \sqrt{\frac{B\kappa^2}{Su_c^2}}, \quad \beta = \sqrt{\frac{\Sigma u_c}{B\kappa}}. \tag{48}$$

so that the parameters $\alpha$ and $\beta$ are related to the characteristic length scales defined in **Equation 46** through

$$\alpha = \left( \frac{\ell_\beta}{\ell_c} \right)^2, \quad \beta = \frac{\ell_c}{\ell_\alpha}, \tag{49}$$

where $\ell_c = \sqrt{u_c/\kappa}$ is a length scale corresponding to the geometric average of the longitudinal radius of curvature and the critical extension $u_c$.

The condition **Equation (47)** can be reformulated most conveniently as

$$\beta > \frac{\sqrt{1-\alpha}}{\alpha} \quad \Leftrightarrow \quad \alpha = \frac{\sqrt{1+4\beta^2}-1}{2\beta^2}, \tag{50}$$

The resulting curve is shown in a phase diagram in **Appendix 2—figure 4a**.

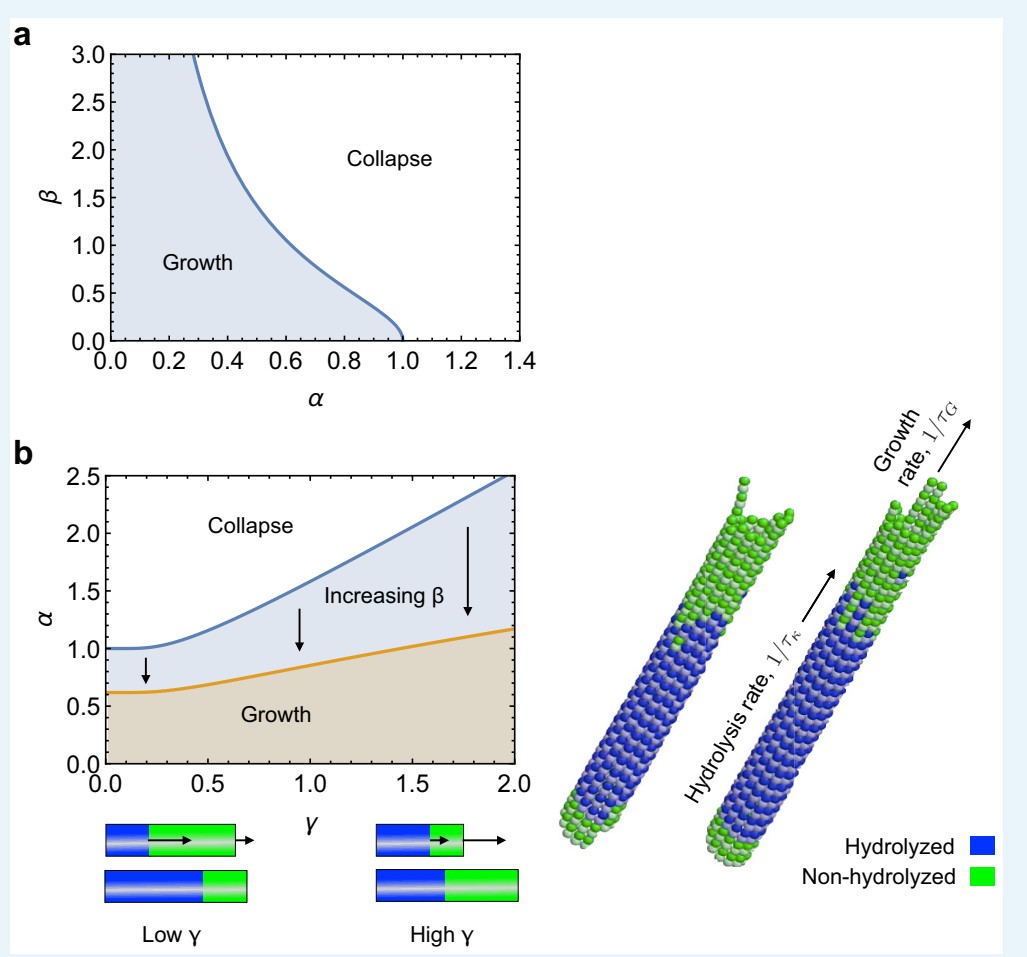

**Appendix 2—figure 2.** Role of kinetics on mechanical stabillity of MTs. (**a**) Phase diagram for mechanical stability of 3D MTs, as described by *Equation 50*. (**b**) Effect of the relative rates of tubulin hydrolysis and MT growth ($\gamma = k_G/k_H$) on mechanical stability of MTs, as described by *Equation 62*. See: link to *Videos 1* and *2*.

## Scaling behaviour of critical curvature with mechanical parameters

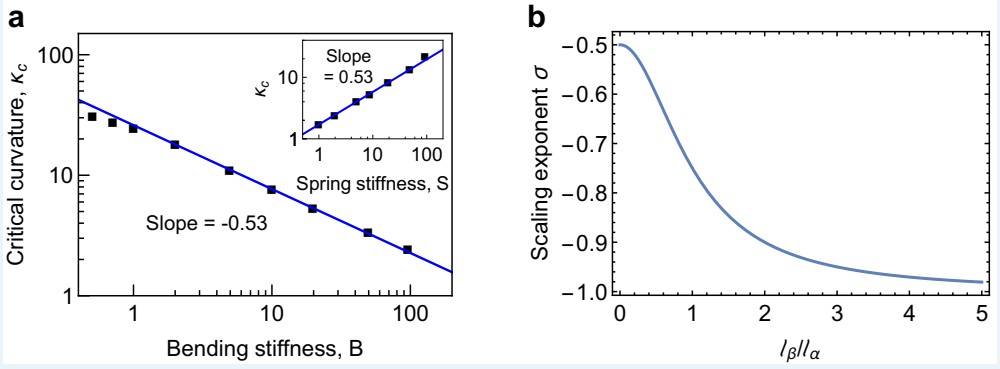

**Appendix 2—figure 3.** Scaling behaviour of critical curvature. (**a**) Scaling behavior of the critical longitudinal curvature $\kappa_c$ with bending stiffness $B$ and spring stiffness $S$ (inset). The scaling relationship over the range of $B$ and $S$ values investigated in the simulations is found to be

$\kappa_c \propto (B/S)^{-0.53}$, in agreement with the predictions of **Equation 55**. (b) Scaling exponent $\sigma$ (**Equation 55**) as a function of $\ell_\beta/\ell_\alpha$ interpolates between the limits $\sigma = -0.5$ for $\ell_\beta/\ell_\alpha \to 0$ and $\sigma = -1$ for $\ell_\beta/\ell_\alpha \to \infty$.

The online version of this article includes the following source data is available for figure 3:

**Appendix 2—figure 3—source data 1.** This spreadsheet contains the data for **Appendix 2—figure 3a**.

**Equation (50)** predicts that there is a critical value for the longitudinal curvature, $\kappa_c$, above which MTs are mechanically unstable. We now study the scaling behaviour of $\kappa_c$ with system parameters such as the bending stiffness $B$ and spring stiffness $S$. To this end, we performed a series of computer simulations by varying the longitudinal bending stiffness $B$ of PFs and the lateral connecting strength $\epsilon$, which in turn determines the spring stiffness $S$. The results are shown in **Appendix 2—figure 3a**. We find that $\kappa_c$ follows a scaling law with the longitudinal bending stiffness $B$ and $S$, $\kappa_c \sim (B/S)^\sigma$. This scaling behaviour is shown in the double-logarithmic plots in **Appendix 2—figure 3a**, where the slope corresponds to the scaling exponent $\sigma \sim -0.53$. The observed scaling behaviour can be rationalised using (50), which can be rewritten as

$$\kappa_c \propto (B/S)^\sigma, \tag{51}$$

where $\sigma$ is the scaling exponent, given by

$$\sigma = \frac{\partial \log(\kappa_c)}{\partial \log(B/S)}. \tag{52}$$

After rewriting **Equation 47** as

$$\kappa_c = \frac{u_c}{\ell_\beta^2 \left(1 + \frac{\ell_\beta^2}{\ell_\alpha^2}\right)} \tag{53}$$

we find

$$\sigma = \frac{\partial}{\partial \log(B/S)} \left[ \log(u_c) - \log\left(\ell_\beta^2\right) - \log\left(1 + \frac{\ell_\beta^2}{\ell_\alpha^2}\right) \right] = -\frac{1}{2} - \frac{1}{\left(1 + \frac{\ell_\beta^2}{\ell_\alpha^2}\right)} \frac{\partial}{\partial \log(B/S)} \left(\frac{\ell_\beta^2}{\ell_\alpha^2}\right), \tag{54}$$

where in the second step we have used $\ell_\beta = (B/S)^{1/4}$ (**Equation 46**). Finally, using $\partial x^a / \partial \log x = a x^a$, we arrive at:

$$\sigma = -\frac{1}{2}\left(1 + \frac{1}{1 + \frac{\ell_\alpha^2}{\ell_\beta^2}}\right) = \begin{cases} -0.5 & when\ \ell_\beta \ll \ell_\alpha \\ -1 & when\ \ell_\beta \gg \ell_\alpha \end{cases} \tag{55}$$

The scaling exponent $\sigma$ interpolates between $-0.5$ for $\ell_\beta \ll \ell_\alpha$ (limit of a single-filament model of MT **Zapperi and Mahadevan, 2011**) and $-1$ for $\ell_\beta \gg \ell_\alpha$ (limit of strong lateral curvature), see **Appendix 2—figure 4b**. This scaling behaviour for $\kappa_c$ has been verified using our coarse grained simulations of depolymerising MTs in **Appendix 2—figure 3a** of the main text. Representative values for the mechanical parameters in the simulations are $B = 1.23 \times 10^{-26}$ Nm$^2$, $S = 91$ MPa and $R = 10$ nm. These values give $\ell_\beta = (B/S)^{1/4} \simeq 3.4$ nm, $\ell_\alpha = 10$ nm and, therefore, $\sigma \simeq -0.55$, in close agreement with the simulations in **Appendix 2—figure 4a**.

## Coupling mechanical stability with MT kinetics

We now combine our static calculation of mechanical stability of MTs with the dynamics of subunit hydrolysis and MT polymerisation/depolymerisation. This allows us study how kinetics affect the phase diagram of **Appendix 2—figure 3a**. Let $k_G$ be the rate of addition of subunits to the MT end and let $k_H$ be the rate of hydrolysis of GTP-tubulin dimers. With these

parameters we can construct a dynamic dimensionless parameter as the ratio between the times for hydrolysis and growth

$$\gamma = \frac{k_G}{k_H}.$$ (56)

To understand how the dynamic parameter $\gamma$ modifies the static stability of the MTs, our starting point are the kinetic equations in the main text:

$$\frac{d\kappa(t)}{dt} = k_H(\kappa(\infty) - \kappa(t)), \quad \frac{d\omega(t)}{dt} = k_H(\omega(\infty) - \omega(t)),$$ (57)

$$\frac{dn(t)}{dt} = k_+[m] - k_- = k_G.$$ (58)

**Equations 57 and 58** can be combined together by expressing time as $t = n/k_G$ (follows directly from the solution to **Equation 58**). The solution to **Equation 57** can thus be expressed as

$$\kappa(n) = \kappa\left(1 - e^{-k_H n/k_G}\right) + \kappa(0)e^{-k_H n/k_G} = \kappa(\infty)\left(1 - e^{-n/\gamma}\right) + \kappa(0)e^{-n/\gamma},$$ (59)

$$\omega(n) = \omega(\infty)\left(1 - e^{-n/\gamma}\right) + \omega(0)e^{-n/\gamma}.$$ (60)

Using this parametrisation, we can express the condition for mechanical instability, **Equation 47**, as

$$\left(1 - e^{-n/\gamma}\right)\alpha\left[1 + \alpha\beta^2\left(1 + \frac{\omega_0 R}{1 - \omega_0 R}e^{-n/\gamma}\right)^2\right] > 1.$$ (61)

This condition can be solved with respect to $\alpha$ to yield the following expression for the phase boundary:

$$\alpha = \frac{\sqrt{1 + \frac{4\beta^2\left(1 + \frac{\omega_0 R}{1 - \omega_0 R}e^{-n/\gamma}\right)^2}{1 - e^{-n/\gamma}}} - 1}{2\beta^2\left(1 + \frac{\omega_0 R}{1 - \omega_0 R}e^{-n/\gamma}\right)^2}.$$ (62)

The resulting phase diagram as a function of $\gamma$ is shown in **Appendix 2—figure 4b**. We see that increasing $\gamma$ increases the region of mechanical stability of MTs. The physical interpretation of this result is that MTs remain mechanically stable as long as the transition curve, **Equation 47**, is reached after a stabilising cap of length $n$ is added onto the MT end. Increasing the rate of MT growth, $k_G$, over subunit hydrolysis, $k_H$, favours the formation of the stabilising cap. In the limit when $\omega$ does not vary with time, **Equation 61** reduces to

$$\left(1 - e^{-n/\gamma}\right)\alpha(1 + \alpha\beta^2) > 1,$$ (63)

which can be solved to yield **Equation 13** of the main text.

## Mechanical stability of MTs in the presence of laterally-distributed quenched disorder

As argued in the main text, in the presence of lateral disorder, the mechanical stability of a MT is controlled by its weakest link. To understand this quantitatively, we replace the MT with lateral disorder with an equivalent one with uniform spring stiffness $\langle S_{\min} \rangle$, where $\langle S_{\min} \rangle$ denotes the average value of the weakest spring stiffness. This replacement maps the study of the mechanical stability of a MT with lateral disorder onto a problem of extreme value statistics (**Zapperi and Mahadevan, 2011**; **Bertalan et al., 2014a**): the determination of

$\langle S_{\min} \rangle$. To this end, consider $N$ independent and identically distributed links with spring constants $S_1, \cdots, S_N$. We assume that the values of spring constants are random and distributed according to a Gamma-distribution

$$p(S) = \frac{k^k S^{k-1} \exp(-kS/\langle S \rangle)}{\langle S \rangle^k \Gamma(k)}, \tag{64}$$

where $\langle S \rangle$ is the average spring stiffness (**Appendix 2—figure 4a**). The ratio between the standard deviation $\sigma$ and the average $\langle S \rangle$

$$C_v = \frac{\sigma}{\langle S \rangle} = \frac{1}{k} \tag{65}$$

is the coefficient of variation, a key parameter which we use to describe the degree of disorder in the system; small values of $k$ correspond to nearly ordered system while large values correspond to a strongly disordered system.

With $S_{\min} = \min_i S_i$ being the smallest value of spring constants and letting

$$P_N(S) = Pr[S_{\min} < S] \tag{66}$$

be the cumulative probability distribution for $S_{\min}$ over the $N$ links, we can calculate $P_N(S)$ directly from the definition of $P_N(S)$, yielding

$$1 - P_N(S) = Pr[S_{\min} \geq S] = \prod_{i=1}^{N} Pr[S_i \geq S] = (1 - P(S))^N, \tag{67}$$

where

$$P(S) = \int_0^S p(S')dS' \tag{68}$$

is the cumulative probability distribution of $S$. For large $N$, we can approximate the exact expression in **Equation 67** as

$$P_N(S) = 1 - (1 - P(S))^N \simeq 1 - \exp[-NP(S)]. \tag{69}$$

The interesting behaviour of $P_N(S)$ happens for small values of $S$, when $P_N(S)$ is controlled by the low-value tail of $P(S)$. The cumulative distribution function $P(S)$ can then be calculated explicitly from **Equation 64** as

$$P(S) = \frac{\gamma(k, kS/\langle S \rangle)}{\Gamma(k)}, \tag{70}$$

where $\gamma(k,x)$ is the incomplete gamma function. Since we are interested in the low-value tail of $P(S)$, we can use the small $x$ expansion of $\gamma(k,x)$

$$\gamma(k,x) \simeq \frac{x^k}{k} + \cdots, \quad x \to 0 \tag{71}$$

to arrive at the following expression

$$P(S) = \frac{\gamma(k, kS/\langle S \rangle)}{\Gamma(k)} \simeq \frac{(kS/\langle S \rangle)^k}{\Gamma(k+1)}, \tag{72}$$

which is valid for $S \to 0$ (**Appendix 2—figure 4b**). The cumulative probability distribution for $S_{\min}$ then converges to a Weibull distribution

$$P_N(S_{\min}) = 1 - \exp\left[-N(S_{\min}/S_0)^k\right], with S_0 = \frac{\Gamma(k+1)^{1/k}\langle S \rangle}{k}. \tag{73}$$

The average value for the weakest spring constant $S_{\min}$ is therefore

$$\frac{\langle S_{\min} \rangle}{\langle S \rangle} = \frac{1}{k} \, \Gamma\left(\frac{k+1}{k}\right) \left[\frac{\Gamma(k+1)}{N}\right]^{1/k}. \tag{74}$$

Having solved the extreme-value statistics problem of determining $\langle S_{\min} \rangle$ allows us to estimate the transition curve separating regions of mechanical stability and instability for a MT exhibiting lateral disorder. The curve separating these zones is obtained by replacing $\langle S \rangle$ by $\langle S_{\min} \rangle$ in *Equation 50*, yielding:

$$\alpha = \frac{\sqrt{1+4\beta^2}-1}{2\beta^2} \sqrt{\frac{\langle S_{\min} \rangle}{\langle S \rangle}}, \tag{75}$$

where we note that $\langle S_{\min} \rangle < \langle S \rangle$. Thus, by comparing *Equation 75* with the deterministic result *Equation 50*, we see that, in the presence of lateral disorder, the curve separating mechanically stable from mechanically unstable MTs shifts in such a way as to increase the region of mechanical instability. Lateral disorder thus weakens MTs, making them more susceptible to catastrophic failure.

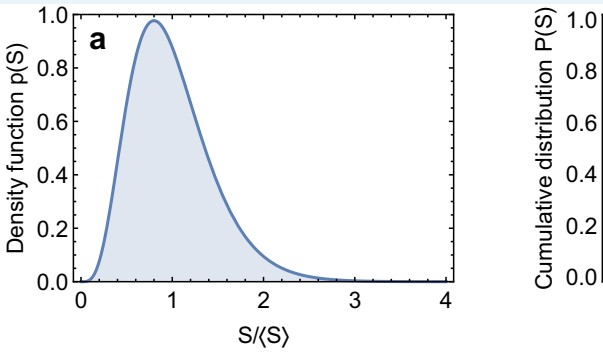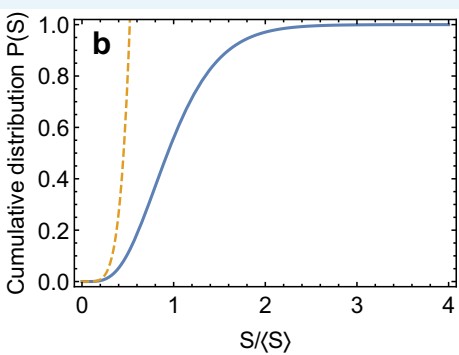

**Appendix 2—figure 4.** Gamma distribution. (**a**) Gamma distribution *Equation 64* with $k = 5$. (**b**) Cumulative probability distribution $P(S)$ for the Gamma distribution, *Equation 70*, and low value-tail, *Equation 72* (dashed line).

