## [Decision Letter]

**Acceptance summary:**

One of the most characteristic features of microtubules is their stochastic growth and shrinkage. The present article provides a microscopic analysis of this process which incorporates the bending elasticity of free ends of filaments that curve away from the microtubule axis along with the effects of disorder in the distribution of hydrolysed tubulin. In bringing together ideas from mechanics and statistical mechanics this work provides new insights into microtubule dynamics.

**Decision letter after peer review:**

[Editors' note: the authors submitted for reconsideration following the decision after peer review. What follows is the decision letter after the first round of review.]

Thank you for submitting your work entitled "Mechanics and kinetics of dynamic instability" for consideration by *eLife*. Your article has been reviewed by two peer reviewers, and the evaluation has been overseen by a Reviewing Editor and a Senior Editor. The following individual involved in review of your submission has agreed to reveal their identity: Jennifer Ross (Reviewer #1).

Both reviewers found the work to be interesting and solid, but they raised

a number of issues regarding the novelty of the research, its connection with

previous work, its predictive nature, and its presentation. Based on consultation between the reviewers and editors, we regret to inform you that your work in its present form will not be considered further for publication in *eLife*. Please see below for the individual reviews. We would in principle be prepared to consider a new submission if the concerns of the reviewers are fully addressed. The paper and the point-by-point rebuttal would then most likely be sent to the same reviewers.

*Reviewer #1:*

The manuscript "Mechanics and kinetics of dynamic instability" from Michaels et al., is a theoretical study of the fundamental causes behind microtubule dynamic instability. The manuscript specifically focuses on the effect of GTP-islands, locations within the lattice where the tubulin has not fully hydrolyzed yet. They use the assumption that the lateral and longitudinal inter-tubulin binding interactions and the conformation of the tubulin dimers both depend on the GTP-hydrolysis state of the tubulin dimer to control the stability of the microtubule structure. Overall, I found the model to be clear (although laced with too much undefined physics jargon) and interesting.

The positive aspects of the manuscript are:

1) The model is based on prior experimental and theoretical results that all seem reasonable.

2) They bring in the concept of frozen-in (quenched) states where the GTP-hydrolysis is stuck in some state to allow the presence of GTP-islands that have been experimentally observed.

3) They also use the concept of percolation (connectedness) of these stable sites to understand why the phase change from growth to shrinkage would be so dramatically sensitive to the size and shape of the GTP-island.

Some problematic aspects of the manuscript:

1) The model put forward is fairly derivative as it is based on prior theory and models. It is entirely possible that I am mistaken, but if so, the authors should explain the novelty of their model, which is not well-explained. What does this model bring to the table that prior models have not? This will help readers and people who might want to use it know when it would be appropriate to use or compare to their experiments.

2) The model suggests at the end that it could make predictions, but it actually fails to make those predictions. Actual, specific predictions would make the manuscript far more useful to experimentalists. For instance, what if you have a 9-protofilament microtubule? How sensitive is it to protofilament number? Can it tell the difference between 9 and 10 protofilament microtubules? In the same field of view?

3) The model does not explain some important aspects of dynamic instability. Specifically, it does not explain the switch from growing phase to shrinkage phase. Or if it does, that was not articulated well. It does a better job of explaining rescue, but with the limitation that the GTP-islands are frozen states. Although it is nice to confirm that this is true with a model, the model is not as realistic as it should be because it is unlikely that the GTP-islands are actually frozen-in. More likely, there is a rate of GTP turn-over that is statistical and probabilistic, so the locations of the islands are dynamic and tend to break the connectedness (percolation) as time proceeds. I understand the importance of initial toy models, but this model, especially given the inarticulation of specific predictions, could do so much more.*Reviewer #2:*The manuscript by Michaels and coauthors develops theory and simulations of the dynamic instability of microtubules. The authors use two approaches: (1) a coarse-grained simulation model of a tubulin dimer with variable curvature and interacting patches to represent lateral and longitudinal interactions and (2) a continuum mechanical model of a microtubule as an elastic sheet. The authors use these models to show several results. First, they demonstrate that stronger lateral interactions and lower longitudinal interactions stabilize microtubules (favor growing over shrinking). Second, they introduce, without physical motivation, random variation in the spring constant of longitudinal and lateral interactions, and show that random variation of lateral interactions destabilizes microtubules (as expected, because breaking of lateral bonds is required for shrinking), but random weakening of longitudinal interactions strengthens microtubules (again as expected, because a single strong ring can keep the microtubule together). Third, the authors consider how hypothesized GTP islands could allow rescue, by introducing into the model GTP islands with varying properties into their simulations and studying the rescue probability. They find that the GTP island length and the GTP fraction of dimers in the island must be high enough for GTP tubulin to percolate the island to achieve a high probability of rescue.

While this paper is technically sound, in my opinion it does not meet the novelty standard of *eLife*. The results are exactly what one would intuitively expect, so the paper reads as using mathematics to demonstrate previously expected results on dynamic instability. The nature of the modeling and the results obtained are similar to many previously found. Previous related work is not cited, including Zakharov et al., 2015. In particular, the model in this manuscript is quite similar to that of Zakharov et al., so what new insight is gained from the current work is unclear.

Additionally, this manuscript does not cite recent work suggesting that tubulin dimers and protofilaments have intrinsic curvature in the GTP-bound state, which would change the nature of the model and possibly the conclusions.

In addition, the authors have not made sufficient effort to use language consistent with that used in the field, or to explain technical terms from other fields. For examples, the authors refer to the rams' horns of shrinking MTs as a flower-like shape. Similarly, the authors use the term quenched disorder without explaining clearly that it simply means random variation in a property of the model, which will confuse readers who are not already familiar with the condensed-matter usage of quenched disorder.

Therefore I recommend that the paper be published in a more specialized journal (such as the Biophysical Journal, which would be a good fit for this paper) and significantly rewrite the paper to clarify the presentation and the connection to previous work.

[Editors' note: further revisions were suggested prior to acceptance, as described below.]

Thank you for submitting your article "Mechanics and kinetics of dynamic instability" for consideration by *eLife*. Your revised article has been reviewed by

one of the original reviewers, and the evaluation has been overseen by a Reviewing Editor and Detlef Weigel as the Senior Editor. The following individual involved in review of your submission has agreed to reveal their identity: Jennifer Ross (Reviewer #1).

As you can see from the review below, there are only minor changes needed to the manuscript in order for it to be acceptable for publication. Please make these at your earliest convenience.

*Reviewer #1:*

The manuscript "Mechanics and Kinetics of Dynamics Instability" by Michaels, et al., is a revision of a prior submission to *eLife*. The goal of the manuscript is to understand the mechanisms of dynamic instability of the 3D model. After having read the new version and the response to reviews, I feel the manuscript is a good contribution to the literature. In particular, I would like to commend the authors on their ability to communicate the physical concepts in much clearer terms. It is clear that they spent time and effort to make the manuscript more accessible. In so doing, they made the manuscript more impactful. It was well-worth the effort. Thank you also for clarifying the predictions of this model.

The authors state that, "MT growth… is powered by the hydrolysis of GTP." This is not true. Dynamic instability is powered by GTP hydrolysis, since the hydrolysis is required for depolymerization, but growth simply needs the GTP binding – not hydrolysis. Perhaps this sentence, which is a compound sentence, meant something else, when it said, "'it' is powered by the hydrolysis of GTP." Whatever, the way it reads currently is that the authors are stating that growth is powered by GTP hydrolysis, which is opposite to currently accepted mechanisms.

---

## [Author Response]

[Editors' note: the authors resubmitted a revised version of the paper for consideration. What follows is the authors' response to the first round of review.]

Reviewer #1:… Some problematic aspects of the manuscript:1) The model put forward is fairly derivative as it is based on prior theory and models. It is entirely possible that I am mistaken, but if so, the authors should explain the novelty of their model, which is not well-explained. What does this model bring to the table that prior models have not? This will help readers and people who might want to use it know when it would be appropriate to use or compare to their experiments.

We would like to respectfully disagree with the statement that our model is fairly derivative. Previous theoretical work on the mechanics of dynamic instability has been almost exclusively computational, with the number of parameters often exceeding the number of predictions. The few studies that have combined theory and computer simulations to model dynamic instability mechanically (e.g. Zapperi, LM 2011) have been limited to considering MTs as one-dimensional chains. To the best of our knowledge, our work is the first to bring together mechanics and kinetics of dynamic instability within a 3D setting which is not incremental.

Accounting for the full 3D geometry of MTs fundamentally changes the predictions relative to the 1D models qualitatively. In 1D, the presence of quenched disorder at the level of GTP-hydrolysis always stabilises MTs. In a 3D setting the quenched disorder acts to destabilise MTs. This difference has profound consequences on the model predictions regarding the stabilising role of GTP remnants. Within a 1D theory, all remnants would be stabilising, i.e. we would predict that the rescue frequency of such remnants would be equal to 1. This is however not what is observed in experiments. Our theory suggests that this might be related to the 3D nature of the remnants, as captured by our percolation model, which we may be the first to introduce to the study of this problem. Our resulting phase diagram which combines two geometric parameters and one kinetic parameter summarizes the role of the tubular structure, the kinetics of hydrolysis and the role of disorder is new, but perhaps this did not come through in our earlier version. We have taken the reviewer's criticism to heart and have revised the main text to better explain the key contributions of our work (see e.g. revised Abstract and Introduction).

2) The model suggests at the end that it could make predictions, but it actually fails to make those predictions. Actual, specific predictions would make the manuscript far more useful to experimentalists. For instance, what if you have a 9-protofilament microtubule? How sensitive is it to protofilament number? Can it tell the difference between 9 and 10 protofilament microtubules? In the same field of view?

These are good suggestions. It may be possible for example to measure unbinding velocities of MT at constant temperature as a function of MT radius or number of PFs. A direct prediction from our theory is that unbinding velocity increases with decreasing number of PFs. The dependence is exponential (through the thermal energy); as such, differences in unbinding velocities might be sufficiently large to be detected experimentally. We have included a paragraph describing our predictions for catastrophe as a function of MT radius in the revised manuscript (see subsection 'Condition for mechanical stability').

3) The model does not explain some important aspects of dynamic instability. Specifically, it does not explain the switch from growing phase to shrinkage phase. Or if it does, that was not articulated well. It does a better job of explaining rescue, but with the limitation that the GTP-islands are frozen states. Although it is nice to confirm that this is true with a model, the model is not as realistic as it should be because it is unlikely that the GTP-islands are actually frozen-in. More likely, there is a rate of GTP turn-over that is statistical and probabilistic, so the locations of the islands are dynamic and tend to break the connectedness (percolation) as time proceeds. I understand the importance of initial toy models, but this model, especially given the inarticulation of specific predictions, could do so much more.

As discussed in the response to the previous comment, we can use our model to account for the switching to depolymerisation. MTs in the mechanically unstable region of the phase diagram in Figure 3 undergo catastrophe. MTs below the stability line can undergo instability in the presence of thermal fluctuations. The catastrophe rate depends exponentially the 'distance' from the stability line, which represents an activation energy barrier. Following the reviewer's comment, we have articulated this explicitly in the revised manuscript (subsection 'Condition for mechanical stability'). Regarding the choice to model GTP using quenched disorder, it is important to appreciate that the term 'frozen'does not imply static. Instead, it means 'slowly-varying' compared to some underlying fast dynamics. In the context of MTs, the use of slowly hydrolyzable analogues of GTP-tubulin contributes to rescue (Tropini et al., 2012). This suggests that the fraction of GTP-tubulin in remnants is random but its composition varies slowly with time, compared to the fast kinetics of polymerisation and depolymerization. Mathematically, this means that there is a separation of timescales between fast polymerization/depolymerization and the slow variation of island properties. According to the framework of non-linear dynamics/asymptotic analysis we can consider polymerization/depolymerization kinetics separately from the island kinetics (in the non-linear dynamics language this is termed a 'slow manifold'). In practice, the parameters describing island composition can be simply replaced by time dependent ones, which makes our theory applicable to the situation highlighted by the reviewer. As long as disorder varies slowly compared to the characteristic timescale of polymerization/depolymerization, one can make the amount of disorder a function of time. We have now made this point explicit in the main text (subsections 'Mechanical stability of 3D MTs in the presence of quenched disorder' and 'Computer simulations').

Reviewer #2:... While this paper is technically sound, in my opinion it does not meet the novelty standard of eLife. The results are exactly what one would intuitively expect, so the paper reads as using mathematics to demonstrate previously expected results on dynamic instability. The nature of the modeling and the results obtained are similar to many previously found. Previous related work is not cited, including Zakharov et al., 2015. In particular, the model in this manuscript is quite similar to that of Zakharov et al., so what new insight is gained from the current work is unclear.Additionally, this manuscript does not cite recent work suggesting that tubulin dimers and protofilaments have intrinsic curvature in the GTP-bound state, which would change the nature of the model and possibly the conclusions.

We thank the reviewer for giving us the opportunity to better explain the novelty of our results. Previous theoretical work on the mechanics of dynamic instability has been almost exclusively computational. The few studies that have combined theory and computer simulations to model dynamic instability mechanically have been limited to considering MTs as one-dimensional chains. To the best of our knowledge, our work is the first to bring together mechanics and kinetics of dynamic instability within a 3D setting which is not incremental.

Accounting for the full 3D geometry of MTs fundamentally changes the predictions relative to the 1D models qualitatively. In 1D, the presence of quenched disorder at the level of GTP-hydrolysis always stabilises MTs. In a 3D setting the quenched disorder acts to destabilize MTs. This difference has profound consequences on the model predictions regarding the stabilizing role of GTP remnants. Within a 1D theory, all remnants would be stabilizing, i.e. we would predict that the rescue frequency of such remnants would be equal to 1. This is however not what is observed in experiments. Our theory suggests that this might be related to the 3D nature of the remnants, as captured by our percolation model, which we may be the first to introduce to the study of this problem. Our resulting phase diagram which combines two geometric parameters and one kinetic parameter summarizes the role of the tubular structure, the kinetics of hydrolysis and the role of disorder is new, but perhaps this did not come through in our earlier version. We have taken the reviewer's criticism to heart and have revised the main text to better explain the key contributions of our work. and used mathematics to translate these results into testable predictions.

In this context, we have extended the Discussion at the end, which now includes a discussion of explicit predictions that could be tested experimentally. We also thank the reviewer for pointing out the reference Zakharov et al., 2015, which is indeed relevant to our work and we have now appropriately cited in the revised manuscript. The key difference between our work and Zakharov et al. is that our work provides an analytical theory, while Zakharov et al. is a simulational study. Moreover, Zakharov et al. studies the MT tip, while our work we have focused on reinforcing islands. There are however important similarities. In particular, Zakharov et al. studied the role disorder at the level of the MT tip composition varies over long timescales compared to growth/shrinkage; this situation is entirely analogous to our consideration of quenched disorder at the level of the reinforcing island's composition. We have also followed the reviewer's suggestion to cite further recent literature on the curvature of the tubulin dimer in its GTP- and GDP-bound forms. In this context, we would like to point out that our model already accounts for a slight intrinsic curvature of the GTP-tubulin dimer (see Table 1); we have clarified this point in the revised manuscript.

In addition, the authors have not made sufficient effort to use language consistent with that used in the field, or to explain technical terms from other fields. For examples, the authors refer to the rams' horns of shrinking MTs as a flower-like shape. Similarly, the authors use the term quenched disorder without explaining clearly that it simply means random variation in a property of the model, which will confuse readers who are not already familiar with the condensed-matter usage of quenched disorder.Therefore I recommend that the paper be published in a more specialized journal (such as the Biophysical Journal, which would be a good fit for this paper) and significantly rewrite the paper to clarify the presentation and the connection to previous work.

We have taken the reviewer's comment to heart and have put significant effort to revise our manuscript and improve accessibility. In particular, we have made sure to use technical language consistent with the MT field; moreover, we have explained technical terms such as, for example, quenched disorder (see Section 'Mechanical stability of 3D MTs in the presence of quenched disorder') to make the paper more accessible to an interdisciplinary readership that might not be familiar with the physics literature (see also comment of reviewer 1). We hope that the revised manuscript has clarified these points.

[Editors' note: what follows is the authors' response to the second round of review.]

Reviewer #1:... The authors state that, "MT growth... is powered by the hydrolysis of GTP." This is not true. Dynamic instability is powered by GTP hydrolysis, since the hydrolysis is required for depolymerization, but growth simply needs the GTP binding – not hydrolysis. Perhaps this sentence, which is a compound sentence, meant something else, when it said, "'it' is powered by the hydrolysis of GTP." Whatever, the way it reads currently is that the authors are stating that growth is powered by GTP hydrolysis, which is opposite to currently accepted mechanisms.

We thank the reviewer for raising this comment. We have rephrased the sentence accordingly (see the Introduction).